# A single-cell multiomics roadmap of zebrafish spermatogenesis reveals regulatory principles of male germline formation

Ana María Burgos-Ruiz [1], Fan-Suo Geng[2], Gala Pujol [3,4], Estefanía Sanabria[1], Thirsa Brethouwer [1], María Almuedo-Castillo[1], Aurora Ruiz-Herrera [3,4], Juan J Tena [1✉] & Ozren Bogdanovic [1,2,5✉]

## Abstract

Spermatogenesis is the biological process by which male sperm cells (spermatozoa) are produced in the testes. Beyond facilitating the transmission of genetic information, spermatogenesis also provides a potential framework for inter- and transgenerational inheritance of gene-regulatory states. While extensively studied in mammals, our understanding of spermatogenesis in anamniotes remains limited. Here we present a comprehensive single-cell multiomics resource, combining single-cell RNA sequencing (scRNA-seq) and single-cell chromatin accessibility (scATAC-seq) profiling, with base-resolution DNA methylome (WGBS) analysis of sorted germ cell populations from zebrafish (*Danio rerio*) testes. We identify the major germ cell types involved in zebrafish spermatogenesis as well as key drivers associated with these transcriptional states. Moreover, we describe localised DNA methylation changes associated with spermatocyte populations, as well as local and global changes in chromatin accessibility leading to chromatin compaction in spermatids. Notably, we identify loci that evade global chromatin compaction, and which remain accessible, suggesting a potential mechanism for the intergenerational transmission of gene-regulatory states. In summary, this high-resolution atlas of zebrafish spermatogenesis provides a valuable resource for studying vertebrate germ cell development and epigenetic inheritance, while offering a robust framework for comparative analyses across diverse models of germ cell biology.

**Keywords** Spermatogenesis; Zebrafish; Epigenetics; Single-cell Multiomics; DNA Methylation
**Subject Category** Development

## Introduction

Spermatogenesis is a biological process, which in animals occurs continuously throughout adult reproductive life. During spermatogenesis, mature haploid sperm cells (spermatozoa) are produced through a series of differentiation events from diploid spermatogonial stem cells (de Kretser et al, 1998). While it was initially believed that sperm solely provided its genetic material to the egg, myriad recent studies have revealed that environmental factors, such as poor diet, exposure to toxins, and stress, can disrupt gene-regulatory marks in sperm thereby influencing offspring traits (Argaw-Denboba et al, 2024; Boskovic and Rando, 2018; Lismer and Kimmins, 2023; Siklenka et al, 2015; Skvortsova et al, 2018). Understanding the process of spermatogenesis and its regulatory principles is thus of crucial importance for a thorough understanding of animal reproductive processes and infertility, with applications in assisted reproductive technologies, animal breeding, and diagnostics of diverse genetic and potentially epigenetic conditions. Single-cell sequencing technologies have enabled the generation of precise high-resolution atlases of human, mice, and primate spermatogenesis revealing both convergent and divergent pathways of spermatogenesis, each defined by their own markers, meiotic regulators, and specific ratios of germ cell populations (Green et al, 2018; Guo et al, 2018; Guo et al, 2020; Guo et al, 2021; Nie et al, 2022; Shami et al, 2020). Additionally, recent epigenome profiling studies have revealed that transcriptional changes during mammalian spermatogenesis are by and large paralleled by changes in chromatin accessibility (Huang et al, 2023), 3D genome structure (Alavattam et al, 2019; Vara et al, 2019), and that the initiation of meiosis is characterised by global DNA demethylation (Siebert-Kuss et al, 2024). Importantly, this global DNA methylome reprogramming event observed in spermatocytes was also observed during murine spermatogenesis (Huang et al, 2023), thus indicating a degree of evolutionary conservation in spermatogenesis-associated chromatin regulation. To date, the majority of genomic and imaging data of non-mammalian vertebrate (anamniote) spermatogenesis have been generated in the zebrafish (*Danio rerio*) teleost model (Leal et al, 2009; Schulz et al, 2010; Ye et al, 2023).

---

[1]Centro Andaluz de Biología del Desarrollo, CSIC-Universidad Pablo de Olavide-Junta de Andalucía, Seville, Spain. [2]Garvan Institute of Medical Research, Sydney, NSW, Australia. [3]Institut de Biotecnologia i Biomedicina, Universitat Autònoma de Barcelona, Cerdanyola del Vallès, Spain. [4]Departament de Biologia Cel·lular, Fisiologia i Immunologia, Universitat Autònoma de Barcelona, Cerdanyola del Vallès, Spain. [5]School of Biotechnology and Biomolecular Sciences, University of New South Wales, Sydney, NSW, Australia. ✉E-mail: juan.tena@csic.es; o.bogdanovic@csic.es

Mammalian and zebrafish spermatogenesis share core processes but exhibit differences in regulation, cellular dynamics, and organisation, the most important of which is the anatomy of the testis itself. In zebrafish, testes are organised into discrete cysts where spermatogenesis occurs synchronously within each cyst (Yoshida, 2016). In mammals, on the other hand, spermatogenesis occurs in seminiferous tubules with radial organisation, whereas germ cells are organised in layers that progress from spermatogonia located near the basal membrane to mature sperm in the lumen (Griswold, 2016). Other major differences observed in spermatogenesis between zebrafish and mammals include modes of paracrine and endocrine signalling, spermatogenesis duration, and sperm anatomy. A recent single-cell RNA sequencing (scRNA-seq) study conducted on a single biological replicate provided a useful snapshot of germ cell populations present in the zebrafish testis (Qian et al, 2022), whereas other high-resolution genomics work described how aging (Sposato et al, 2024) and environmental factors (Haimbaugh et al, 2022; Pujol et al, 2025) impact on zebrafish spermatogenesis. However, detailed genome-scale descriptions of the regulatory dynamics of anamniote spermatogenesis are still lacking, hindering our full understanding of this crucial process. To generate high-resolution transcriptional and gene-regulatory atlases of zebrafish spermatogenesis, we employed scRNA-seq and single-cell chromatin accessibility (scATAC-seq) profiling in biological replicates and combined these datasets with low-input whole-genome bisulfite sequencing (WGBS) of germ cell populations sorted from zebrafish testes. In our work, we provide a detailed atlas of zebrafish male germ cell types, each characterised by multiple novel transcriptional drivers. Furthermore, we identify localised DNA methylation remodelling in spermatocytes as well as transitions in chromatin accessibility leading to global chromatin compaction during spermatogenesis. Finally, we characterise in detail the chromatin makeup of elongated spermatids, thus providing insight into loci with potential for intergenerational transmission of gene-regulatory states.

## Results

### Cell-type resolved transcriptomes of the zebrafish testis

To generate comprehensive transcriptional and regulatory maps of zebrafish spermatogenesis, we obtained single-cell suspensions from testes of adult males ($n = 2$) and prepared scRNA-seq and scATAC-seq libraries compatible with the 10x Genomics platform (Fig. 1A; Dataset EV1; Appendix Fig. S1). For RNA-seq, we sequenced a total of 8,432 cells from two biological replicates (Fig. EV1A,B). After initial filtering for the number of expressed genes and percentage of mitochondrial reads, both replicates displayed comparable count profiles and unsupervised cluster numbers. Most cells belonged to germ cell populations, however, small populations of somatic cells ($n = 125$) (Appendix Fig. S2A,B), comprising Leydig (expressed markers: *insl3*, *cyp11a1*, *cyp17*, *star*, *hsd3bl*, and others) (Tremblay, 2015) (Appendix Fig. S2C; Dataset EV2), Sertoli (expressed markers: *krt18a.1*, *aldh1a2*, *fxyd6*, *olfml3*, *cxcl12* and others) (De Gendt et al, 2014; Gilbert et al, 2009; Qian et al, 2022; Raverdeau et al, 2012) (Appendix Fig. S3 and Dataset EV2), and hematopoietic immune cells (expressed markers:

*rac2*, *coro1a*, *grap2* and others) (Deng et al, 2011; Li et al, 2012; Ma et al, 2001) (Appendix Fig. S4; Dataset EV2), were also identified. Interestingly, a subpopulation of Sertoli cells displayed specific expression of genes such as *gstt1a*, *gpx3*, *uraha*, *aqp3*, *gsdf* and others, as revealed by UMAP feature plots (Appendix Fig. S3B). These genes are associated with oxidative stress response, solute transport, and signalling functions, suggesting that this subset may represent a metabolically specialised or stress-responsive state. Alternatively, the divergence might reflect spatial heterogeneity within the testis, or a temporary functional programme related to germ cell interaction. Overall, the identified proportion of somatic cells (1.8%) is in line with previous zebrafish studies (Qian et al, 2022; Sposato et al, 2024); however, we acknowledge that this is likely an underrepresentation due to technical limitations associated with cell size and tissue dissociation, as noted previously (Sposato et al, 2024). Given the high concordance of the two datasets, after excluding clusters exclusive to a single replicate and selecting only germ cell clusters, we integrated both replicates to obtain a total of 6755 cells (Fig. 1B). After manual curation of the clusters, which was based on expression of previously defined marker genes (Qian et al, 2022; Ye et al, 2023), we annotated major germ cell populations: undifferentiated spermatogonia-A (SPG-Aun), differentiated spermatogonia-A (SPG-Ad), spermatogonia B (SPG-B), primary spermatocytes (SPC-I), secondary spermatocytes (SPC-II), round spermatids (SPT-r), and elongated spermatids (SPT-e) (Figs. 1B,C and EV1C). Given the critical role of spermatogonia in maintaining stem cell potential and initiating spermatogenic commitment, we next focused on delineating potential differentiation trajectories within this compartment. To this end, we employed RNA velocity analysis (La Manno et al, 2018) to infer both the direction and magnitude of predicted transcriptional progression across SPG populations. In our analysis, we observed prominent velocity vectors extending from undifferentiated spermatogonia (SPG-Aun) toward differentiated spermatogonia (SPG-Ad), aligning with the expected direction of developmental commitment. Notably, a subset of SPG-Ad cells exhibited velocity vectors oriented towards the undifferentiated state (SPG-Aun) (Appendix Fig. S5), suggesting transcriptional plasticity and potential reversibility within this compartment. These observations indicate that both undifferentiated and differentiated SPG populations may contribute to the emergence of SPG-B cells, reflecting a metastable and uncommitted cellular landscape, consistent with the dynamic plasticity previously reported in human spermatogonia populations (Guo et al, 2018).

We next assessed the total number of cells within each population and found comparable representation of major germ cell types across biological replicates (Fig. 1D), consistent with previous observations (Sposato et al, 2024). Nevertheless, these proportions differ from those reported in histological studies (Leal et al, 2009), which typically show spermatogonia as a minor population. The relative enrichment of spermatogonia in our scRNA-seq dataset likely reflects a known technical artefact; during enzymatic or mechanical tissue dissociation and microfluidic capture, more resilient cells, such as spermatogonia, are preferentially recovered, whereas more fragile or structurally embedded cells, such as spermatids, are often underrepresented. This phenomenon has been documented previously (Denisenko et al, 2020) and likely explains the distribution observed in our dataset. Moreover, we observed that the spermatogenesis process is

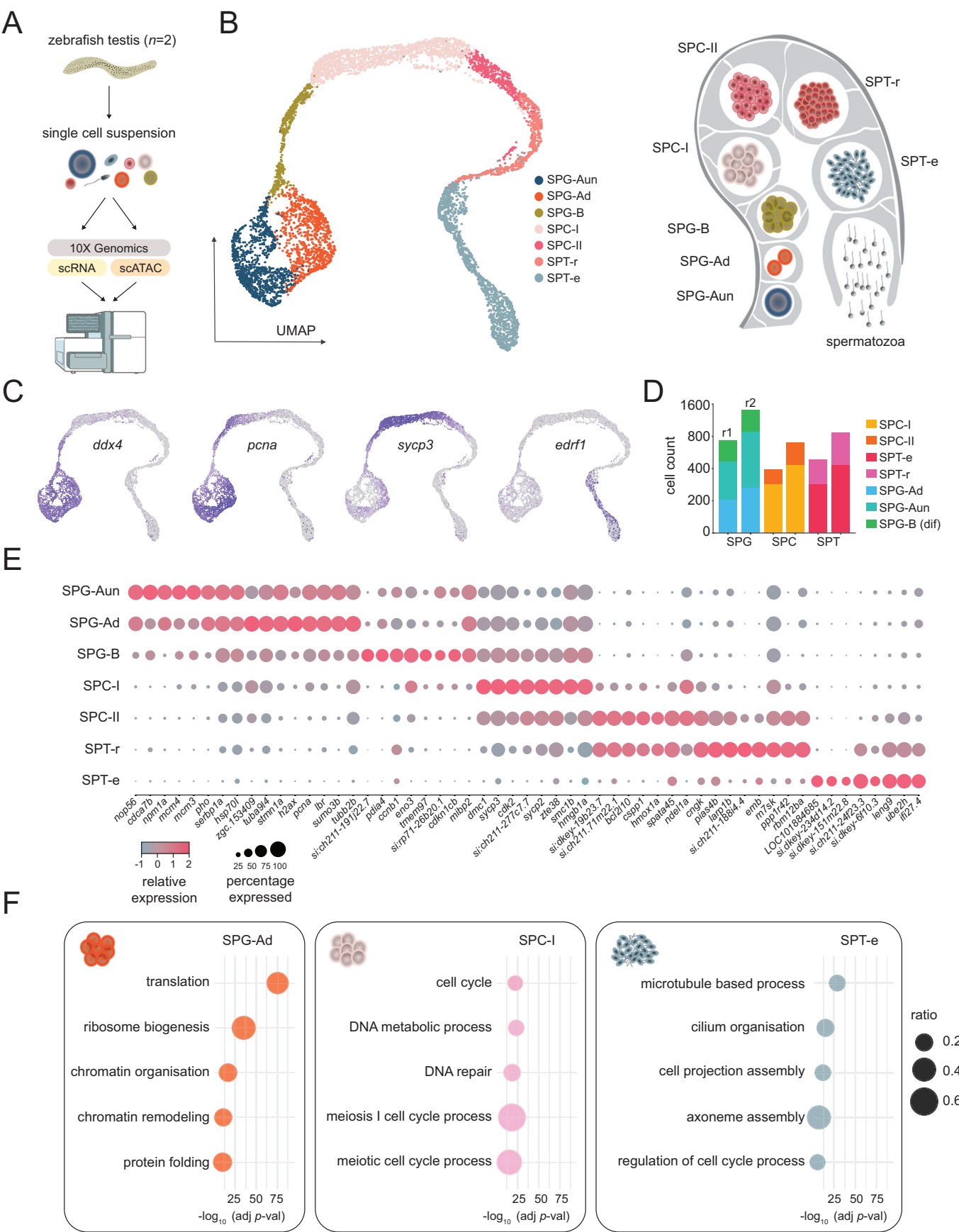

◀ **Figure 1. Single-cell RNA-sequencing (scRNA-seq) of the zebrafish testis.**

(A) Schematic representation of experimental design and sequencing strategy. (B) Left panel: UMAP (uniform manifold approximation and projection) plots of the zebrafish testis tissue with annotated cell types: undifferentiated spermatogonia-A (SPG-Aun), differentiated spermatogonia-A (SPG-Ad), spermatogonia B (SPG-B), primary spermatocytes (SPC-I), secondary spermatocytes (SPC-II), round spermatids (SPT-r), and elongated spermatids (SPT-e). Right panel: schematic drawing of the spermatogenesis process in *Danio rerio*. (C) Examples of marker gene expression: *ddx4* (SPG), *pcna* (proliferation), *sycp3* (SPC), *edrf1* (SPT). (D) Number of cells per cell type across both biological replicates (r1—replicate 1, r2—replicate 2). (E) Average expression level and percentage of cells expressing each marker gene. (F) Most highly significant gene ontology processes associated with marker genes for SPG-Aun, SPC-I, and SPT-e. Log-transformed ($-$log10) adjusted *P* values (FDR) are represented on *x* axes. Enrichment *P* values were computed with Fisher's exact one-tailed test (cumulative hypergeometric). Source data are available online for this figure.

paralleled by a gradual transcriptional shutdown, with elongated spermatids being almost entirely transcriptionally quiescent (Fig. EV1D). To better understand the RNA processing dynamics during spermatogenesis, we first examined the proportion of exonic and intronic reads across germ cell populations (Appendix Fig. S6A). Intronic reads, which primarily represent unspliced pre-mRNA, can serve as a proxy for nascent transcription (La Manno et al, 2018). We found that the ratio of intronic to exonic reads remained relatively constant from spermatogonia to spermatids, suggesting a coordinated downregulation of both transcription and mRNA abundance without substantial accumulation of mature transcripts in later stages. We next quantified the absolute number of intronic reads per cell across biological replicates (Appendix Fig. S6B). This analysis revealed a progressive reduction in intronic read counts from early to late germ cell stages, consistent with a gradual decline in transcriptional activity during spermatid maturation (Fig. EV1D). Together, these findings support a model of transcriptional shutdown that occurs in a stepwise manner, likely involving both repression of transcription initiation and increased transcript turnover.

Following the assessment of transcriptional dynamics, we next focused on the identification of marker genes associated with each cell population. This analysis, based on the newly assigned cell-type identities, besides identifying previously described germline genes, identified dozens of novel cell-type-specific markers of zebrafish spermatogenesis (Fig. 1E; Dataset EV3). We next assessed gene ontology enrichments corresponding to the identified marker genes, which revealed categories in accord with their cellular function (Figs. 1F and EV1E; Dataset EV4). For example, SPG-Aun and SPG-Ad populations were enriched in terms associated with ribosome biogenesis and translation, further supporting the findings that the transition from self-renewal to germline differentiation is dependent on ribosome biogenesis and increased protein synthesis (Kawasaki et al, 2025). SPC-I cells were enriched in terms associated with meiosis and DNA repair (Griswold, 2016), whereas SPT-r and SPT-e populations displayed enrichment in terms linked to cilium assembly (Mirvis et al, 2018). Interestingly, the SPG-Ad population was enriched in categories such as "chromatin remodelling" and "chromatin organisation", indicative of chromatin structure changes taking place during spermatogonial differentiation. Having identified marker genes that recapitulate germ cell-specific patterns of expression, we next wanted to search for major spermatogenesis drivers, genes that exhibit dynamic expression patterns, which reflect temporal or progressive changes during spermatogenesis. To that end, we applied an unsupervised approach for inferring linear developmental chronologies from scRNA-seq data (Cannoodt et al, 2016) and identified 162 genes (Figs. 2A and EV2A; Dataset EV5). Moreover, to validate our driver

gene set we applied an alternative approach for trajectory inference which combines dimensionality reduction with graph-based methods (Cao et al, 2019) (Fig. 2B). This approach resulted in a somewhat less conservative gene population ($n = 608$), 97% of which overlapped our initial driver set (Fig. EV2B). We next proceeded to experimentally validate drivers corresponding to major germ-cell populations by fluorescent in situ hybridisation. We chose three novel drivers: *setb*, *hmgb1b*, and *ckba* that displayed distinct expression profiles corresponding to early, mid, and late spermatogenesis stages, respectively (Fig. 2C and EV2C). All three targets were characterised by strong staining in germ-cells with *setb* signal coinciding with *ddx4*, a canonical spermatogonial marker (Ye et al, 2023), and *ckba* co-staining with *sumo1*, a marker of later spermatogenesis stages (Vigodner and Morris, 2005). Expectedly, *hmgb1b* displayed an intermediate staining profile in line with its expression pattern in spermatocytes (Fig. 2D). Overall, our single-cell transcriptomes of the zebrafish testis provide a comprehensive overview of zebrafish spermatogenesis, delineate novel cell-type-specific driver genes, and provide insight into cellular function.

## Localised DNA methylation changes characterise spermatocyte formation

To provide insight into gene-regulatory processes taking place during spermatogenesis, we next studied DNA methylation (5mCG), a major gene-regulatory mark required for spermatogenesis (Barau et al, 2016; Dura et al, 2022). Moreover, 5mCG is stably maintained throughout the anamniote life cycle thereby offering a potential template for paternal epigenetic inheritance (Jiang et al, 2013; Potok et al, 2013; Ross et al, 2023; Skvortsova et al, 2019). To that end, we generated base resolution DNA methylome (WGBS) datasets from germ cell populations sorted from zebrafish testes (Fig. 3A). We obtained a mix of undifferentiated and differentiated spermatogonia (SPG), SPC-I, SPT-r populations, as well as mature sperm (SP) (Fig. 3B,C). To investigate whether zebrafish spermatogenesis is characterised by global DNA methylome reprogramming processes akin to those observed in eutherian mammals (Huang et al, 2023; Siebert-Kuss et al, 2024), we first assessed global 5mCG levels in these cell populations and found that the DNA methylome is stably maintained throughout spermatogenesis (Fig. 3D). Similarly, we observed strong correlation ($r = 0.94–0.96$), between distinct spermatogenesis stages when 5mCG levels were compared in 10 kb genomic blocks (Figs. 3E and EV3A). To identify genomic regions exhibiting localised changes in 5mCG, we analysed DNA methylome profiles to detect differentially methylated regions (DMRs) > 100 bp long and with a minimum change in the fraction of methylated CpG sites

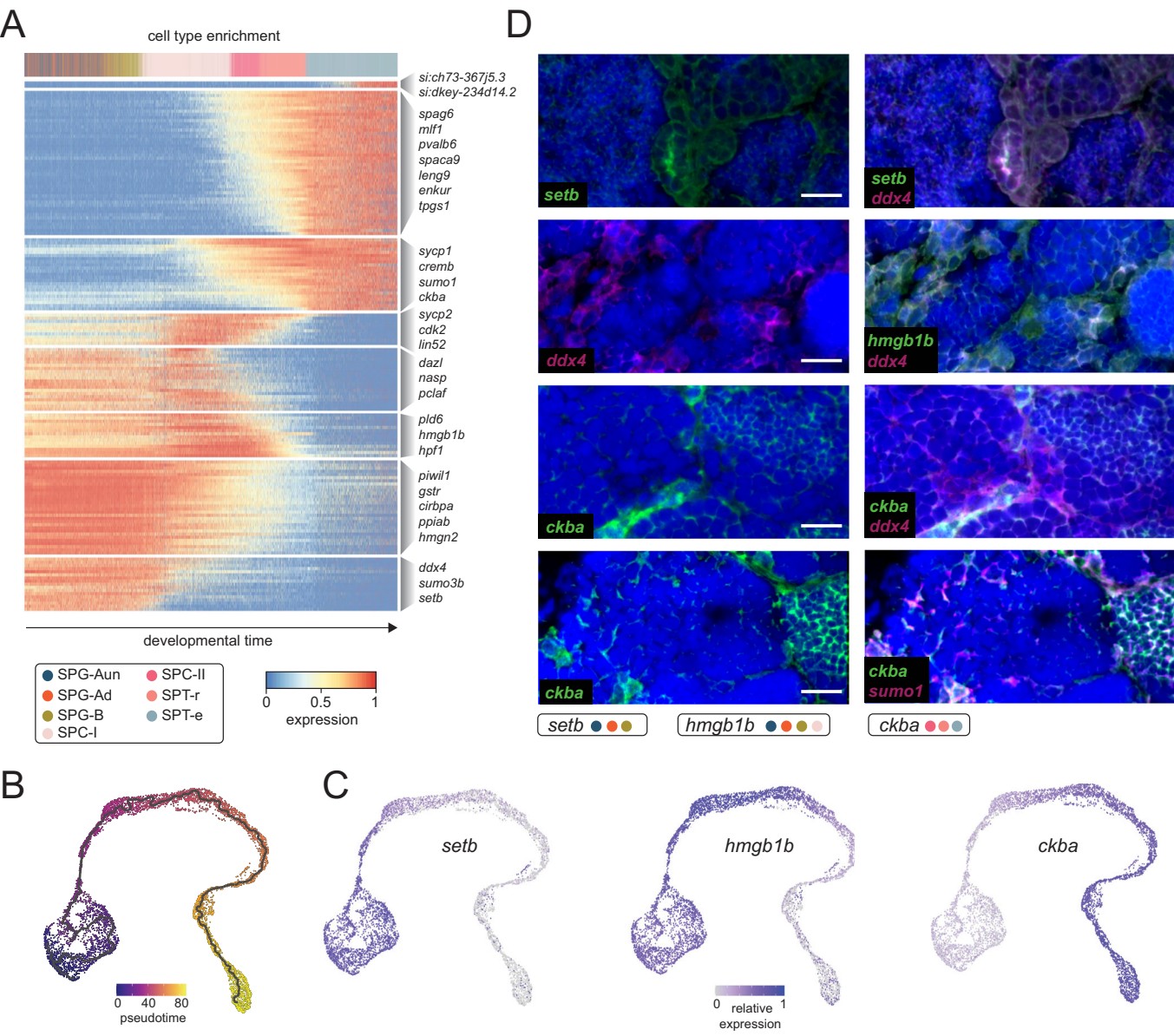

**Figure 2.  Driver genes of zebrafish spermatogenesis.**

(A) Heatmap showing gene ($n = 158$) expression dynamics, with cells ranked according to their position in an inferred trajectory using SCORPIUS. Cells are colour-coded by cell type, with key genes highlighted in clusters corresponding to trajectory-driven modules. (B) UMAP (uniform manifold approximation and projection) plots of the zebrafish testis tissue denoted by pseudotime and trajectory using Monocle. (C) Expression pattern of newly identified spermatogenesis driver genes (*setb*, *hmgb1b* and *ckba*). (D) Double fluorescent in situ validation of *setb*, *hmgb1b* and *ckba* against previously defined markers (*ddx4*, *sumo1*). Scale bars: 10 µm. Source data are available online for this figure.

($\Delta$mCG) of 0.2 ($P < 0.05$, Wald test). This approach revealed 3879 localised changes in DNA methylation occurring during zebrafish spermatogenesis. *K*-means clustering ($k = 4$) revealed different clusters all of which were predominantly characterised by changes in 5mCG states associated with the SPC-I population (Fig. 3F,G). Those involved both hypo- and hypermethylated regions in SPC-I. The identified DMRs likely arise from a combination of active de novo methylation, driven mainly by high expression of *dnmt3bb.2*, and passive demethylation, potentially due to reduced *dnmt1* expression during these stages (Fig. EV3B). The absence of *tet1*, *tet2*

and *tet3* expression in spermatocytes supports the view that active DNA demethylation is unlikely to contribute significantly to methylation dynamics during this stage of spermatogenesis. Instead, changes in 5mCG are more likely driven by a combination of de novo methylation and passive mechanisms. In addition, the differential expression of methyl-CpG binding domain (MBD) proteins suggests stage-specific interpretation of methylation marks; *mbd1*a is predominantly expressed in spermatogonia (SPG-Aun, SPG-Ad and SPG-B), whereas *mbd2* is selectively expressed at later stages, including SPC-II and SPT-r.

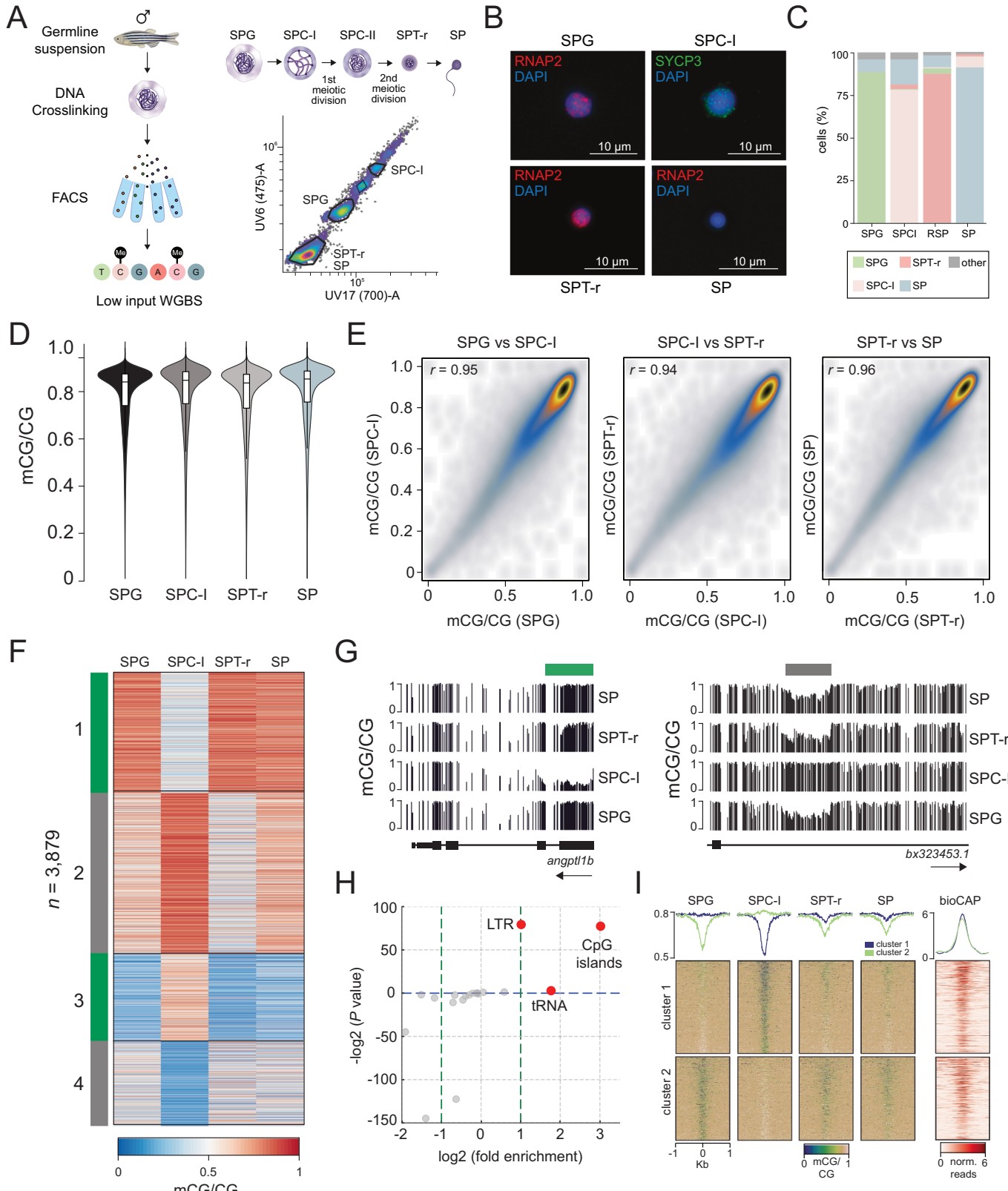

**Figure 3. Base resolution DNA methylomes (WGBS) of zebrafish spermatogenesis.**

(A) Schematic representation of the germ extraction and sequencing protocol and the sorted cell populations (SPG—spermatogonia, SPC-I— spermatocytes I, SPT-r— round spermatids, SP—mature sperm). (B) Representative immunofluorescence images displaying DAPI-stained DNA along with specific proteins for diverse cell populations. DAPI (blue), SYCP3 (green), RNAP2 (red). (C) Percentage of different cell populations per isolated fraction. (D) DNA methylation levels (mCG) in sorted cell populations quantified across non-overlapping 10 Kb genomic bins ($n = 168,917$). The outer shape represents the kernel density estimate, while the boxplot inside highlights the median, interquartile range (IQR), and whiskers. Whiskers extend to the smallest and largest values within 1.5 times the IQR, with points beyond this range considered outliers. (E) Scatter plots of average 5mCG levels in 10 kb genomic bins ($R$ = Pearson correlation coefficient). Shown are: SPG vs SPC-I, SPC-I vs SPT-r, and SPT-r vs SP comparisons. For all comparisons, please refer to Fig. EV3A. (F) K-means ($k = 4$) clustering of DMRs identified between SPG and SPC-I, SPC-I and SPT-r, and SPT-r and SP methylomes. (G) Representative examples of SPC-I hypo- (left panel), and SPC-I hyper-mCG (right panel) DMRs. (H) Enrichment of genomic features associated with DMRs. Points marked in red were deemed statistically significant ($P < 0.05$, hypergeometric test). (I) Positional heatmaps of 5mCG and bioCAP (CpG island enrichment) signal plotted over DMRs. Source data are available online for this figure.

To obtain further insight into the genomic context of 5mCG changes, we annotated the DMRs according to their genomic location (Heinz et al, 2010) (Dataset EV6; Fig. EV3C) and identified a highly significant enrichment in CpG islands (CGIs) (Fig. 3H and Appendix Table S1). CGIs are genomic regions with elevated CpG density and GC content that frequently coincide with vertebrate gene promoters and other gene-regulatory elements (Angeloni and Bogdanovic, 2021; Deaton and Bird, 2011). In agreement with these results, the identified DMRs displayed strong bioCAP (CFP1-CxxC domain enrichment) signal (Data Ref: Long et al, 2013) indicative of their CGI colocalization and their regulatory function (Fig. 3I). Overall, our DNA methylome datasets reveal thousands of localised, bi-directional 5mCG changes at potential gene-regulatory regions occurring during spermatocyte stages, a time window, which in eutherian mammals is characterised by large-scale 5mCG remodelling (Huang et al, 2023).

## Global and local dynamics of chromatin accessibility during spermatogenesis

To study gene-regulatory mechanisms that operate during spermatogenesis, we generated 10x Chromium-compatible scATAC-seq libraries from scRNA-seq-matched testes tissues (Fig. 1A; Dataset EV1; Appendix Fig S1). For scATAC-seq, we sequenced a total of 17,392 single cells from two biological replicates, and after filtering for doublets and low-quality cells (Appendix Table S2), we obtained 2099 cells for replicate one and 3251 cells for replicate two. We processed our scATAC-seq data by normalising it for sequencing depth before applying dimensionality reduction. Using UMAP, we visualised the relationships between cells, revealing distinct populations, which were annotated by computing differential gene activity scores, predicted from the number of ATAC-seq fragments between clusters (Fig. 4A,B; Dataset EV7). After assessing the concordance of biological replicates (Fig. EV4A), we merged the data in order to obtain a single collection of 5350 cells, which was used for downstream analysis. The annotation was further refined by identifying marker genes linked to essential biological processes in spermatogenesis (Fig. EV4B,C). For instance, undifferentiated spermatogonia (SPG-Aun) were classified based on the presence of spermatogonial stem cell markers such as *id4*, *lin28*, and *nanos1* (Helsel et al, 2017; Koprunner et al, 2001; Lord and Nixon, 2020; Zheng et al, 2009). In contrast, differentiated spermatogonia (SPG-d; comprising a mixture of SPG-Ad and SPG-B cells) were identified by the absence of these markers and the activity of genes, including *smad6*, *socs2*, *nop56*, and *zranb2*, which were previously associated with

later stages of spermatogonia and their progression (Guo et al, 2018; Itman and Loveland, 2008; Orwig et al, 2008; Shami et al, 2020). Spermatocytes I (SPC-I) were characterised by high-level activity of genes such as *sycp3*, *esco2*, *majin* and others, which are factors essential for the early stages of meiosis, where their main role is the facilitation of homologous chromosome pairing, synapsis, and sister chromatid cohesion (Shibuya et al, 2015; Syrjanen et al, 2014; Vega et al, 2005). In contrast, spermatocytes II (SPC-II) displayed chromatin opening of canonical spermatocyte markers (*spo11*, *mei4*, *tdrd12*)) (Kumar et al, 2015; Pandey et al, 2013; Romanienko and Camerini-Otero, 2000), mitotic checkpoint genes (*bub3*, *bub1ba*) (Martinez-Exposito et al, 1999), and genes involved in sperm motility (*foxj1* and *cfap20*) (Beckers et al, 2020; Chrystal et al, 2022). Following meiosis, round spermatids (SPT-r) were distinguished by the activity of *dnah1*, *tcte1*, and *izumo1*, which collectively guide the initial steps of spermiogenesis (Ben Khelifa et al, 2014; Castaneda et al, 2017; Inoue et al, 2005). Finally, elongated spermatids (SPT-e) were marked by the upregulation of genes such as *theg*, *iqcg*, and *tssk6*, central to the terminal remodelling events in spermatogenesis (Li et al, 2014; Nayernia et al, 1999; Sosnik et al, 2009).

To understand the extent to which chromatin accessibility changes are paralleled by changes in transcription during spermatogenesis, we selected a subset of markers ($n = 343$) that displayed highly significant changes in chromatin accessibility (log2FC > 0.8 and $P$ val adj < 0.005, MAST test) (Finak et al, 2015) (Fig. 4C) and queried their transcriptional profiles. We observed a clear shift in transcriptional states coinciding with the SPC-I population that was characterised by the shutdown of spermatogonial markers and an increased expression of genes implicated in spermatocyte and spermatid formation. Having completed cluster annotation, we next studied global patterns of chromatin accessibility during spermatogenesis to better understand how global chromatin structure correlated with observed transcriptional and epigenetic changes. We found that the number of total accessible fragments per cell increased progressively from undifferentiated spermatogonial stem cells to differentiated spermatogonia (Figs. 4D and EV4D). This trend continued through meiosis, with peak numbers rising in spermatocytes I and II, indicating robust transcriptional activity during these stages. However, the total number of accessible fragments declined in round and elongated spermatids, consistent with global transcriptional shutdown and chromatin condensation. These findings are consistent with the global pattern of chromatin accessibility changes observed through single-cell approaches during human spermatogenesis (Wu et al, 2022) (Fig. 4E). Overall, our single-cell chromatin accessibility

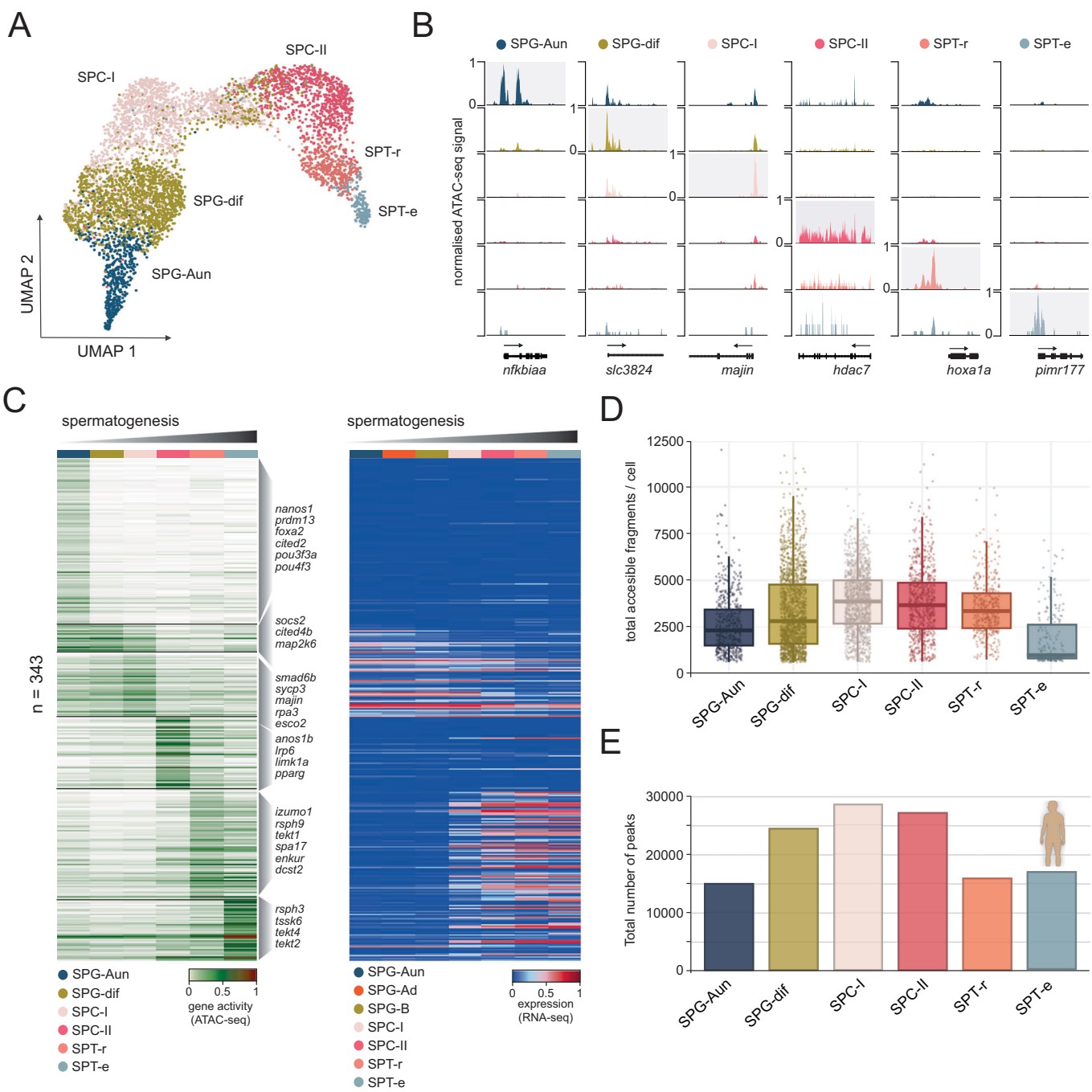

**Figure 4. Single-cell chromatin accessibility dynamics during zebrafish spermatogenesis.**

(A) UMAP (uniform manifold approximation and projection) plots of scATAC-seq data obtained zebrafish testis tissue. The annotated cell types are: undifferentiated spermatogonia-A (SPG-Aun), differentiated spermatogonia A and B (SPG), primary spermatocytes (SPC-I), secondary spermatocytes (SPC-II), round spermatids (SPT-r), and elongated spermatids (SPT-e). (B) Genomic examples of stage- and locus-specific chromatin accessibility across identified cell populations. (C) Heatmaps of gene activity calculated from scATAC-seq data (left panel) and matching scRNA-seq data (right panel), for genes identified as differentially active across clusters based on scATAC-seq data (log2FC > 0.8, adjusted $P < 0.005$). Differential activity was assessed using the MAST test (Model-based Analysis of Single-cell Transcriptomics), which employs a hurdle model to account for the sparsity and dropout typical of single-cell data. (D) Boxplots showing the distribution of total accessible fragments detected in scATAC-seq data. Boxes indicate the interquartile range (IQR), horizontal lines denote the median, whiskers extend to 1.5× IQR, and individual points represent single cells. Sample sizes were: $n = 5350$ cells in total, comprising SPG-Aun ($n = 584$), SPG-dif ($n = 1825$), SPC-I ($n = 1458$), SPC-II ($n = 819$), SPD-r ($n = 423$), and SPD-e ($n = 241$). (E) Total number of ATAC-seq peaks from comparable human populations identified through scATAC-seq profiling (Wu et al, 2022). Source data are available online for this figure.

datasets reveal local chromatin changes associated with activity of spermatogenesis marker genes, as well as gradual reprogramming of global chromatin structure leading to chromatin compaction and transcriptional shutdown.

## Open chromatin peaks coincide with sites of multivalent chromatin in sperm

In zebrafish sperm, "placeholder" nucleosomes and multivalent chromatin states mark thousands of regulatory regions, potentially maintaining them in a poised configuration for rapid activation in the embryo during zygotic genome activation (ZGA) (Murphy et al, 2018). While it is believed that these specialised chromatin features could confer inherited epigenetic information and facilitate early developmental processes, it is not yet clear to what extent these regions remain in an open conformation, indicative of TF binding. To address this, we employed scATAC-seq to examine the chromatin accessibility profiles of elongated spermatids, which are expected to exhibit predominantly condensed chromatin. We identified 2023 ATAC-seq peaks (Dataset EV8) in the SPT-e population, suggestive of some maintenance of open chromatin states during later stages of spermatogenesis. Genes associated with these peaks were enriched for several diverse biological processes, including "chromatin regulation", "cell cycle", and "embryonic development" (Fig. EV5A). We next analysed the transcription factor (TF) binding motifs enriched at sites of open chromatin in elongated spermatids and found a strong association with CGI-bound (NF-Y) (Oldfield et al, 2019) and methylation-sensitive (SP1, ZBTB14, NRF1, YY2) (Cawley et al, 2004; Deaton and Bird, 2011; Gupta et al, 2023) TFs (Fig. 5A; Dataset EV9). Notably, this pattern was conserved throughout all spermatogenesis stages (Fig. EV5B), with most enriched motifs containing a CpG site, which are otherwise strongly depleted in vertebrate genomes (Gardiner-Garden and Frommer, 1987). Importantly, many of these broadly expressed factors (Appendix Fig. S7) (Tapial et al, 2017), as well as other key components of CGI chromatin, were strongly expressed in mature sperm and zebrafish testis tissues, based on RNA-seq data re-analysed from previous studies (Data Ref: Jiang et al, 2013; Data Ref: Valcarce et al, 2023) (Fig. 5B). To clarify the relationship between ATAC-seq signal and DNA hypomethylation, a common feature of CGIs, we de novo identified hypomethylated CpG-rich regions (unmethylated regions – UMRs) (Burger et al, 2013) using our germ cell stage-specific DNA methylation data (Fig. 3). These analyses revealed similar numbers of UMRs throughout spermatogenesis (spermatogonia = 18,406; mature sperm = 18,058) (Fig. 5C), in line with the absence of any major DNA methylome reprogramming events (Fig. 3). Next, we plotted ATAC-seq signal from our single-cell data across mature sperm UMRs, revealing a chromatin reprogramming pattern analogous to the one observed on the genome-wide scale (Figs. 5D and EV5C). Notably, peak intensity and width both varied, with SPG-dif and SPC-I peaks being the broadest. Given the strong correlation between ATAC-seq signal and DNA hypomethylation (Fig. 5D,E), we next asked whether these sites overlap with multivalent "placeholder" chromatin regions (enriched in H3K4me1, H3K4me3, H2AZ, H3K14ac, and hypomethylation) (Fig. 5F) (Data Ref: Murphy et al, 2018). Indeed, late SPT-e ATAC-seq signal strongly coincided with placeholder chromatin, previously identified as a major driver of maternal-to-paternal

chromatin remodelling before ZGA. Taken together, our findings suggest that thousands of open chromatin regions might be retained intergenerationally within the context of placeholder chromatin, likely to facilitate ZGA and early embryonic development.

## Discussion

The process of spermatogenesis is largely conserved among vertebrates and is generally divided into three phases: spermatogonial proliferation, meiosis, and post-meiotic maturation. Numerous studies, including recent single-cell genomics reports, have greatly advanced our understanding of the cellular population dynamics and gene-regulatory events that occur during spermatogenesis (Green et al, 2018; Guo et al, 2021; Huang et al, 2023; Murat et al, 2023; Nie et al, 2022; Shami et al, 2020). However, most of this research was conducted on mammalian models, leaving a significant gap in our knowledge of the regulatory mechanisms governing non-mammalian (anamniote) sperm formation. This is important to appreciate because mammals and anamniotes exhibit major differences in chromatin regulation during germline and early development. For example, mammalian sperm formation involves near-complete replacement of histones with protamines, resulting in highly condensed sperm chromatin, with only a small fraction of histones (1–10%) retained. Nevertheless, the extent of this phenomenon as well as the exact genomic locations of retained loci and their function, remain a topic of debate (Carone et al, 2014; Erkek et al, 2013; Hammoud et al, 2009; Samans et al, 2014; Yin et al, 2023). In contrast, fish display diverse sperm chromatin packaging strategies. Zebrafish, for instance, predominantly use histones to package their sperm (Wu et al, 2011), whereas medaka employ protamine-based packaging (Hong et al, 2004), similar to mammals. The presence of nucleosomes in vertebrate sperm suggests a potential mechanism for intergenerational epigenetic marking, which may be even more pronounced in species like zebrafish that rely on nucleosome-based sperm packaging.

To investigate the gene-regulatory processes involved in zebrafish sperm formation and to better understand the potential for paternal inheritance of epigenetic states, we generated scRNA-seq and scATAC-seq datasets of the zebrafish testis. Our scRNA-seq data revealed seven major germ-cell populations corresponding to spermatogonia (SPG-Aun, SPG-Ad, SPG-B), spermatocytes (SPC-I, SPC-II), and spermatids (SPT-r, SPT-e) (Fig. 1), as well as Leydig, Sertoli and hematopoietic immune cells (Fig. EV1B; Appendix Figs. 1–4). Overall, our findings are consistent with previously published zebrafish scRNA-seq datasets, though we observed minor discrepancies in cell population ratios (Qian et al, 2022), likely due to differences in sequencing depth or zebrafish age (Sposato et al, 2024; Zhang et al, 2020). Consistent with previous work, we also found that the number of expressed genes decreases progressively as spermatogenesis advances (Zhang et al, 2020), resulting in global transcriptional downregulation in elongated spermatids. Finally, by inferring developmental trajectories, we identified novel driver genes across all stages (Fig. 2), generating a resource that will be of great value for understanding infertility and other disease states linked to spermatogenesis. Our open chromatin profiling via scATAC-seq broadly recapitulated the spermatogenesis stages identified through expression profiling, while providing a

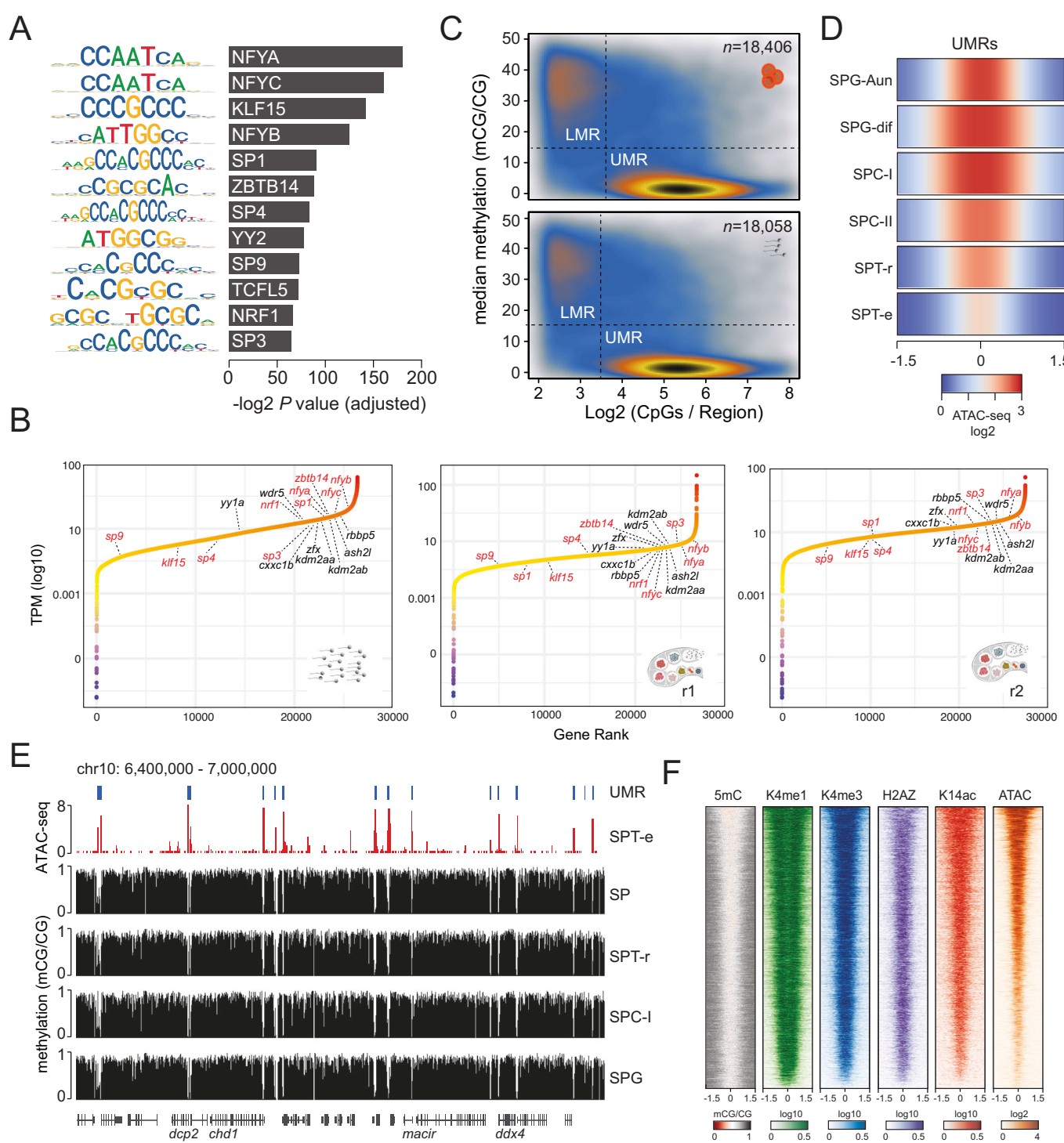

higher-resolution view of gene-regulatory changes (Fig. 4). Differential analysis of scATAC-seq data revealed thousands of changes in chromatin accessibility, many of which paralleled changes in steady-state RNA levels. We further observed a gradual increase in chromatin accessibility during spermatogonial differentiation, peaking at SPC-I and SPC-II stages, followed by a progressive decrease in round and elongated spermatids. This pattern is consistent with Gene Ontology (GO) enrichments of marker genes

expressed in differentiating spermatogonia, which are strongly associated with functions related to chromatin remodelling and organization (Fig. 1). A similar global pattern of chromatin reprogramming was also reported during human spermatogenesis in a recent scATAC-seq study (Wu et al, 2022), whereas scCOOL-seq profiling of human sperm at single-cell resolution revealed a somewhat different pattern yet still exhibited open chromatin peaks during spermatocyte stages (Huang et al, 2023).

**Figure 5.  Chromatin state of elongated spermatids (SPT-e).**

(A) Top enriched TF binding motifs in open chromatin peaks retained in SPT-e, ranked in descending order ($P$ value $< 0.05$; hypergeometric test. (B) Gene expression (transcripts per million, TPM) plotted against gene rank, in transcriptomes of mature zebrafish sperm (left panel) (Data Ref: Jiang et al, 2013) and two biological replicates of zebrafish testis (middle and right panel) (Data Ref: Valcarce et al, 2023). Transcripts coding for TFs that display enriched motifs in SPT-e are marked in red. Other key components of CGI chromatin are highlighted in black. (C) De novo discovery of regulatory regions (UMRs—unmethylated regions—i.e., CGI promoters; and LMRs— lowly methylated regions—i.e., enhancers). Scatter plot representing CpG density plotted against DNA methylation levels in newly identified UMRs and LMRs. Upper panel: spermatogonial UMRs ($n = 18{,}406$), lower panel: mature sperm UMRs ($n = 18{,}058$). (D) scATAC-seq signal (log2) of germ-cell populations plotted over mature sperm UMRs ($n = 18{,}058$). (E) Genomic example of a 600 kb region surrounding the *ddx4* gene demonstrating the concordance between DNA methylation, UMRs and scATAC-seq signal (SPT-e). (F) Positional heatmaps showing (i) DNA methylation in mature sperm (SP), (ii) ChIP-seq enrichment (log10) for H3K4me1, H3K4me3, H2AZ, and H3K14ac in mature sperm (Murphy et al, 2018), and (iii) scATAC-seq accessibility (log2) in SPT-e cells. Regions are sorted by ATAC-seq signal (highest to lowest) and plotted across UMRs ($n = 10{,}000$). Source data are available online for this figure.

To better understand the epigenomic changes that occur during zebrafish spermatogenesis, we complemented our findings with DNA methylome data obtained by WGBS from four germ cell populations. In mammals, two rounds of genome-wide DNA methylation reprogramming occur; one during pre-implantation development and another during primordial germ cell formation (Hackett et al, 2013; Hill et al, 2018; Lee et al, 2014; Santos et al, 2002; Seisenberger et al, 2012; Smith et al, 2014; Smith et al, 2012; Xu and Xie, 2018). More recently, a major DNA methylation reprogramming event specifically at the spermatocyte stage has also been reported in humans (Huang et al, 2023; Siebert-Kuss et al, 2024). However, no large-scale epigenome rearrangements have been observed in zebrafish to date (Ortega-Recalde et al, 2019; Ross et al, 2023; Skvortsova et al, 2019; Wang et al, 2021), suggesting that parental inheritance of gene-regulatory marks and other epigenetic factors from gametes in anamniotes may play a more significant developmental role. Consistent with previous zebrafish DNA methylome profiling studies, we did not detect any major DNA methylation reprogramming events during spermatogenesis (Fig. 3). Interestingly, the majority of statistically significant DNA methylation changes ($n = 3879$; ~0.13% genome) (Fig. 6A) that we identified, were associated either with hyper- or hypomethylation in spermatocytes, which is when DNA methylation reprogramming takes place during spermatogenesis in mammals. Whether these changes in both mammals and anamniotes reflect genuine regulatory events or are simply byproducts of large-scale chromosomal rearrangements (Liu et al, 2017; Melamed-Bessudo and Levy, 2012), remains an open question. Finally, using our DNA methylome data, we de novo identified CpG-rich unmethylated regions (UMRs; 3% genome) and demonstrated that these sites frequently retain open chromatin and multivalent "placeholder" chromatin (Fig. 6B). These findings provide further evidence for intergenerational inheritance of epigenetic marks in zebrafish (Murphy et al, 2018), suggesting that open chromatin and transcription factor binding likely play a role in this process. Several CGI-associated chromatin components (e.g., NRF1, SP5) have already been implicated in infertility and spermatogenesis defects (Wang et al, 2017; Xu et al, 2022), highlighting the potential relevance of CGI chromatin for intergenerational epigenetic transmission (Molaro et al, 2011), though the full extent of this phenomenon remains to be determined. Future studies using scATAC-seq and WGBS could also elucidate whether the age-associated transcriptional changes observed in zebrafish spermatogonia (Sposato et al, 2024), such as repression of *piwil1*, *e2f5*, and *ube2* genes and aberrant activation of *pou5f3*, *nanog*, and *zp3.2*, are driven by alterations in chromatin accessibility or DNA

methylation at key regulatory loci. Such approaches could clarify whether the observed lineage infidelity and impaired differentiation arise from errors in epigenetic reprogramming or stochastic loss of regulatory fidelity. Our study thus provides a valuable first step toward understanding gene-regulatory dynamics during spermatogenesis in anamniotes and offers a framework for addressing fundamental questions in the regulation of this process.

## Methods

### Reagents and tools table

| Reagent/resource | Reference or source | Identifier or catalogue number |
| --- | --- | --- |
| **Recombinant DNA** | | |
| pBluescript KS+ | Addgene | https://www.addgene.org/vector-database/1949/ |
| **Antibodies** | | |
| Anti-RNA polymerase II CTD repeat YSPTSPS (phospho S5) antibody [4H8] | Abcam | Ab5408 |
| Cy™5 AffiniPure® Goat Anti-Mouse IgG (H + L) | Jackson ImmunoResearch Laboratories | 115-175-166 |
| Anti-SCP3 antibody | Abcam | Ab15093 |
| Fluorescein (FITC) AffiniPure® Goat Anti-Rabbit IgG (H + L) | Jackson ImmunoResearch Laboratories | 111-095-003 |
| **Oligonucleotides and other sequence-based reagents** | | |
| Double fluorescent in situ hybridisation (FISH) sequences | This study | Dataset EV10 |
| **Chemicals, enzymes and other reagents** | | |
| Tricaine methanesulfonate (MS-222) | Sigma | 886-86-2 |
| Pico Methyl-Seq Library Prep Kit | Zymo Research | D5456 |
| Unmethylated Lambda DNA | Promega | D1521 |
| Chromium Single Cell 3′ GEM Library & Gel Bead Kit v3 | 10x Genomics | CG000185 Rev B |
| Chromium Single Cell ATAC Reagent Kit | 10x Genomics | CG000168 Rev D |
| Digoxigenin-11-UTP | Roche | 11209256910 |

| Reagent/resource | Reference or source | Identifier or catalogue number |
|---|---|---|
| Fluorescein-12-dUTP | Roche | 11373242910 |
| T3 RNA Polymerase | Roche | RPOLT3-RO |
| **Software** | | |
| Cell Ranger 7.2.0 | Zheng et al, 2017 | |
| Cell Ranger ATAC 1.2 | Satpathy et al, 2019 | |
| Seurat 5.1.0 | Hao et al, 2024 | |
| Signac 1.14.0 | Stuart et al, 2021 | |
| Harmony 1.0.1 | Korsunsky et al, 2019 | |
| Monocle3 1.2.9 | Cannoodt et al, 2016 | |
| SCORPIUS 1.0.9 | Cao et al, 2019 | |
| scDblFinder 1.8.0 | Germain et al, 2021 | |
| Velocyto 0.17 | La Manno et al, 2018 | |
| Scanpy 1.11.3 | Wolf et al, 2018 | |
| scVelo 0.3.3 | Bergen et al, 2020 | |
| Trimmomatic 0.39 | Bolger et al, 2014 | |
| Bismark 0.22.3 | Krueger and Andrews, 2011 | |
| MethylDackel 0.6.1 | https://github.com/dpryan79/MethylDackel | |
| DSS 2.54.0 | Feng et al, 2014 | |
| ImageJ | Schindelin et al, 2012 | |
| g:Profiler | Reimand et al, 2007 | |

## Zebrafish husbandry and ethics

Zebrafish were maintained in 3-L tanks, with a maximum density of 15 adult fish per tank. Zebrafish experiments were approved by the Garvan Institute of Medical Research Animal Ethics Committee under AEC approval 17/22. All procedures complied with the Australian Code of Practice for the Care and Use of Animals for Scientific Purposes. Animal experiments conducted at CABD have been approved by the Animal Experimentation Ethics Committees at the Pablo de Olavide University and CSIC (license number 02/04/2018/041). Procedures at UAB complied with the animal ethics guidelines approved by the University Animal Experimentation Ethics Committee.

## Zebrafish procedures

Adult male zebrafish (*Danio rerio*, AB/Tübingen), aged 6 months, were prepared and anesthetized with 0.25% tricaine methanesulfonate (MS-222) on ice for 15 min before experimentation. Testes were dissected and rinsed three times with PBS. The samples were then digested in 10 ml of 0.25% trypsin at 37 °C for 15 min, with

gentle pipetting every 3 min to facilitate tissue breakdown. Digestion was halted by adding DMEM supplemented with 10% FBS, and the resulting cell suspension was passed through a 70-μm nylon mesh filter. The cells were then centrifuged at 1000 rpm for 10 min, after which the pellet was resuspended in 1 ml of DMEM (with 10% FBS) and filtered through a 40-μm nylon mesh.

## scRNA-seq and scATAC-seq library construction and sequencing

Expression (scRNA-seq) libraries were prepared using the 10x Chromium Single Cell 3′ GEM Library & Gel Bead Kit v3 and sequenced on an Illumina NovaSeq 6000 platform (S4, 200 bp, PE). Chromatin accessibility (scATAC-seq) libraries were generated using the 10x Chromium Single Cell ATAC Reagent Kit and sequenced on an Illumina NextSeq 550 platform with the High Output Kit v2.5 (150 bp, PE). For both scRNA-seq and scATAC-seq, we obtained an average of 25,000–30,000 paired-end reads per cell.

## Single-cell RNA-Seq read alignment and quantification

Raw reads were demultiplexed and mapped to the zebrafish danRer11 reference transcriptome using the 10x Genomics CellRanger (v7.2.0) pipeline (Zheng et al, 2017). Before downstream QC, CellRanger gene expression algorithm identified 3984 cells in replicate 1 and 5996 cells in replicate 2. Following filtering, these counts were reduced to 2852 cells (replicate 1) and 5996 cells (replicate 2), with median gene counts of 2436 and 2388, respectively.

## Pre-processing, quality filtering, batch integration, and dimensional reduction scRNA-seq

Seurat v5.1.0 (Hao et al, 2024)was used for scRNA-seq pre-processing, quality control, and analysis. Raw count matrices from two 10x Genomics replicates were imported. Cells expressing <200 or >7000 genes or with >5% mitochondrial content were excluded. Data were normalised using *LogNormalize* (scale factor: 10,000), and highly variable features were identified via vst. Replicates were integrated using *harmony::RunHarmony*. To identify cell populations, PCA was performed, and the optimal number of PCs was determined using ElbowPlot, selecting the first 11 PCs. A shared nearest neighbour (SNN) graph was built with *Seurat::FindNeighbors*, followed by clustering via Leiden (resolution = 0.5). Clusters with low UMI and gene counts, ribosomal RNA enrichment, or clusters present only in one replicate, were removed as likely artefacts. In addition, clusters expressing Leydig (*cyp17a1*, *hsd3b1*), Sertoli (*fshr*, *nr5a1*), and peritubular myoid cell markers (*n* = 219 cells) were excluded from further analyses. The analysis was repeated on the remaining cells using 6 PCs, yielding 13 clusters. Cells were visualised using UMAP (*RunUMAP*, 8 PCs).

## Cell classification and marker identification

Marker genes for each cluster were identified using *Seurat::FindAllMarkers*, applying a log fold-change threshold of 0.15 and restricting the analysis to positive markers, while excluding genes expressed in fewer than 10 cells. Clusters were manually annotated

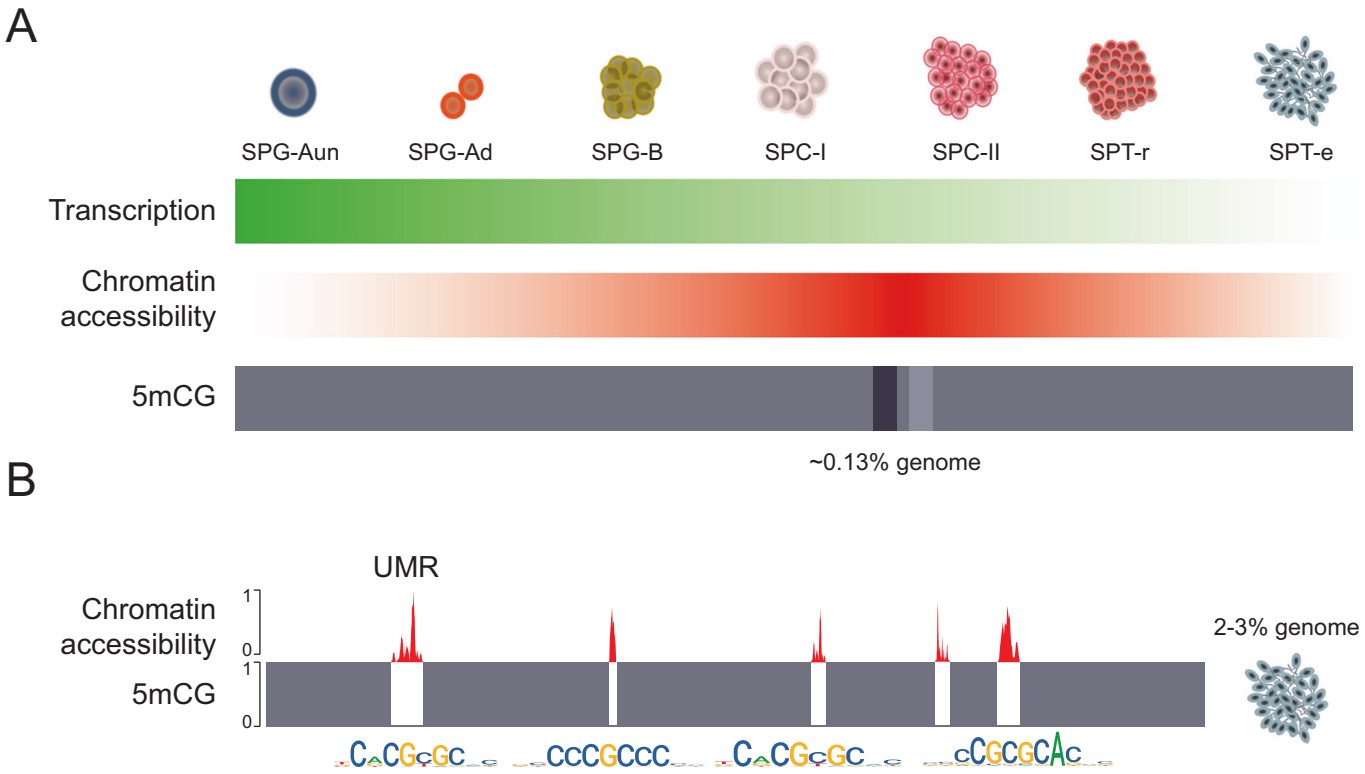

**Figure 6.** Chromatin and transcriptome dynamics during zebrafish spermatogenesis.

(A) Schematic representation of transcriptome (green), chromatin accessibility (red), and DNA methylation (grey) dynamics during zebrafish spermatogenesis. The spermatogenesis transcriptome is characterised by a gradual transcriptional downregulation, leading to the lowest transcriptional activity in elongated spermatids. Chromatin accessibility, on the other hand, gradually increases, reaching peak activity during spermatocyte development (SPC-I and SPC-II). Meanwhile, the DNA methylome remains stable, except for localised methylation and demethylation events (dark and light shading) affecting 0.13% of the genome during spermatocyte formation. (B) Unmethylated regions (UMRs) exhibit a strong ATAC-seq signal and an enrichment in CpG island (CGI) protein-binding motifs in elongated spermatids (SPT-e). This chromatin configuration affects ~2–3% of the genome, and at least a fraction of this epigenetic makeup may be inherited intergenerationally through sperm.

based on literature-defined marker genes (Qian et al, 2022; Ye et al, 2023). Spermatogonia (SPG), which comprised clusters 8, 6, 3, 5, 12, and 10, were characterised by *ddx4* and *piwil1* expression. SPG-A (clusters 8, 6, 3, 5) exhibited higher *ddx4* and *piwil1* expression compared to SPG-B (clusters 10, 12). Within SPG-A, clusters 3 and 5 (A - undifferentiated) were characterised by *eno3*, *e2f5*, and *ripply2*, while clusters 6 and 8 (A - differentiated) expressed *hist1h2a6*. Spermatocytes (SPC), which comprised clusters 0, 4, and 9, were identified by *sycp3* and *pcna* expression, where SPC-I (clusters 0, 4) exhibited high *sycp3* and *pcna* levels and SPC-II (cluster 9) exhibited lower expression of these markers. Spermatids (SPT) were categorised into round SPT (clusters 7, 11), which lacked *edrf1*, and elongated SPT (clusters 2, 1), which expressed *edrf1*. Upon annotation, we conducted a refined analysis by employing a higher log fold-change threshold (0.25), which led to the identification of new zebrafish cell-specific markers, thus further improving cell-type resolution.

### Gene set enrichment analysis

Gene enrichment analysis was conducted using the marker genes identified for each cell type. These genes were input into the g:Profiler website tool (Reimand et al, 2007), with *Danio rerio*

specified as the reference organism. The analysis focused exclusively on enriched biological processes associated with each cell type. The most representative terms were ranked by statistical significance, and the top five biological processes with the lowest adjusted *P* values were selected for visual representation.

### scRNA-seq trajectory analysis

For trajectory analysis of scRNA-seq data, we employed Monocle3 (v1.2.9) (Cannoodt et al, 2016) and SCORPIUS (v1.0.9) (Cao et al, 2019) R packages with imported clustering information from Seurat, including UMAP coordinates and cluster identities. The trajectory graph was constructed using the *monocle3::learn_graph* function, and cells were ordered along the pseudotime trajectory with *order_cells*. Differential expression analysis across pseudotime was performed using *monocle3::graph_test*. SCORPIUS was used to validate the findings, starting with dimensionality reduction using multidimensional scaling (MDS) in three dimensions, based on Spearman correlation distance. A trajectory was inferred with default parameters, and candidate marker genes were identified using *SCORPIUS::gene_importances* function, selecting the top 200 genes for further analysis. Genes were grouped into modules, excluding ribosomal-associated modules (module < 9). To refine

the results, driver genes identified by Monocle3 and SCORPIUS were filtered and compared. For Monocle3, genes were filtered by Moran's I > 0.5 and q value < 0.05, resulting in 604 driver genes. For SCORPIUS, the 162 genes identified through the *SCORPIUS::gene importances* function were extracted. A comparison between the two methods revealed an overlap of 158 genes.

## RNA velocity analysis

Raw 10× Genomics data from replicates 1 and 2 were processed with Velocyto v0.17.17 using run10x to generate spliced and unspliced count matrices, which were concatenated and merged using Scanpy v1.11.3. All downstream velocity pre-processing, filtering and normalisation (*scv.pp.filter_and_normalize*), moment computation (*scv.pp.moments*), stochastic velocity modelling (*scv.tl.velocity*), and velocity graph construction (*scv.tl.velocity_graph*), were performed in scVelo v0.3.3 for UMAP-based trajectory inference.

## Double fluorescent in situ hybridisation (FISH) on zebrafish testis cryostat sections

The coding sequences of *ddx4*, *setb*, *hmgb1b*, *ckba* and *sumo1* (Dataset EV10) were synthesised and introduced into pBluescript KS+ plasmid using *NotI* and *XbaI* restriction enzymes. Antisense riboprobes were synthesised using digoxigenin-11-UTP or fluorescein-12-dUTP and T3 RNA polymerase after *SacI*-linearised plasmid digestion. Cryosectioning of samples was performed as previously described (Letelier et al, 2023) followed by an adapted double FISH protocol (Solana, 2018). Sections underwent Proteinase K treatment (10 μg/mL, 30 min, 37 °C) and post-fixation with 4% formaldehyde (30 min, RT). Probe hybridisation occurred at 70 °C for ≥16 h. After post-hybridisation SSC washes, probes were developed using anti-DIG-POD antibody (Merck, 1:150) and incubated overnight (4 °C). Sections were washed (PBS, Borate buffer) before staining with green TSA amplification (50 μg/mL TSA Fluorescein) for 30 min (RT, dark). POD enzyme was quenched (0.1 M Glycine, pH 2.2), followed by anti-Fluorescein-POD antibody incubation (Merck, 1:150) and overnight (4 °C) incubation. After further washes, red TSA amplification (50 μg/mL Red TSA) was applied (30 min, RT, dark). Sections were washed (PBS) and incubated overnight (4 °C) with DAPI (1:5000, Sigma). Confocal imaging was performed using an LSM 880 (Zeiss) and processed with ImageJ (Schindelin et al, 2012).

## Single-cell ATAC pre-processing and quality control

Replicate data (r1, n = 8305 cells; r2, n = 9088 cells) were processed using the Cell Ranger ATAC (v.1.2) count pipeline (danRer11 assembly). The output of the Cell Ranger ATAC pipeline was utilised as the input for downstream analysis using the Signac R package (v1.14.0). A unified peak set was generated using the *GenomicRanges::reduce* function to merge peak coordinates from both datasets, with subsequent filtering based on peak length. Fragment objects were created for each sample using the *Signac::CreateFragmentObject* function, and peak quantification was conducted using the *Signac::FeatureMatrix* function. Cells deemed low-quality were identified and excluded if they had a nucleosome signal score below 1 and a TSS enrichment score below

5. Meanwhile, cells with more than 40% of fragments mapping to peaks and between 500 and 10,000 fragments in peaks were retained for further analysis. To identify potential doublets, scDblFinder R package (v1.8.0) was applied to the dataset using a converted SingleCellExperiment object from Seurat. Only cells classified as "singlet" were retained for downstream analyses. Following filtering, the number of retained cells was n = 2099 (r1) and n = 3251 (r2). After quality filtering, term frequency-inverse document frequency (TF-IDF) normalisation was applied, followed by the selection of relevant features using a threshold (min.cut-off = 0.5). Dimensionality reduction was performed using singular value decomposition (SVD) to compute latent semantic indexing (LSI) embeddings with up to 70 dimensions. To assess the impact of sequencing depth, LSI components were evaluated using *Signac::DepthCor*, and those with strong correlations to sequencing depth were excluded (LSI1 and LSI10 in replicate 1, LSI1 and LSI8 in replicate 2). The remaining components were used for downstream analyses.

## Single-cell ATAC analysis

Integration of both replicates was performed using the *Seurat::FindIntegrationAnchors* and *Seurat::IntegrateEmbeddings* functions, specifying dimensions 2:6 and 9:70 for integration. Clustering was conducted using the *Seurat::FindCluster* function with the smart local moving (SLM) algorithm for modularity optimisation, with a resolution parameter of 1.2 and algorithm set to 1. Clusters that were found exclusively in one of the replicates were removed. Annotation was performed by first calculating gene activity scores using the *Signac::GeneActivity* function. Annotation was further refined by identifying marker genes associated with key biological processes in spermatogenesis. Using these gene activity scores, the RNA data were log-normalised, and differentially expressed genes were identified using the MAST test (Finak et al, 2015), adjusting for RNA count as a latent variable. Genes with an adjusted P < 0.005 and avg_log2FC > 0.8 were considered significant. Motif position frequency matrices (PFMs) from the JASPAR CORE vertebrate collection were obtained using the *Signac::getMatrixSet* function and added with *Signac::AddMotfs*. Enriched motifs were identified with *Signac::FindMotifs* in peaks accessible in ≥10% of elongated spermatids.

## Flow cytometry and cell enrichment analysis

Zebrafish testes were disaggregated following a previously described protocol (Pujol et al, 2025). The cell suspension was fixed in 1% formaldehyde, centrifuged, and stored at −80 °C until use. Frozen samples were then resuspended in PTBG (0.05% Tween-20 in 1× PBS) and immunostained in solution with a mouse anti-RNA polymerase II antibody (#ab5408, Abcam) diluted in PTBG (1:1000) and incubated at 4 °C overnight. The next day, cells were centrifuged for 15 min at 2000 × g, resuspended in PTBG, and incubated for 5 min at 21 °C with an anti-mouse Cy5 (#115-175-166, Jackson ImmunoResearch, 1:1000). After a PTBG wash, cells were stained with 5 μg/ml Hoechst 33342 for 35 min at 21 °C and sorted using a BD FACS Discover S8 Cell Sorter. Four testicular populations (spermatogonia, spermatocytes I, round spermatids, and spermatozoa) were isolated considering their nucleus complexity, ploidy, and RNA polymerase II staining, with between 60,000

and 400,000 cells collected per cell type. Cell enrichment of each flow-sorted population was evaluated by immunofluorescence. Sorted cells were fixed onto slides by incubating with freshly prepared 4% paraformaldehyde solution containing 0.15% Triton X-100 for 2 h at room temperature in a humidified chamber. After air-drying, the slides were washed in 1% Photo-Flo and incubated with the primary antibodies rabbit anti-SYCP3 (#ab15093, Abcam, 1:100) and mouse anti-RNA polymerase II (#ab5408, Abcam, 1:1000) overnight at 4 °C. After incubation, slides were washed twice in PTBG, followed by a 1-hour incubation at 37 °C with the secondary antibodies anti-rabbit FITC (#111-095-003, Jackson ImmunoResearch, 1:200) and anti-mouse Cy5 (#115-175-166, Jackson ImmunoResearch, 1:1000). DNA was counterstained with antifade solution containing 0.1 μg/ml DAPI and stored at −20 °C until use. Stained slides were analysed using an epifluorescence microscope (Axiophot, Zeiss). Spermatogonia (SPG) exhibited a granulated nucleus and were positive for RNA polymerase II; spermatocytes I (SPC-I) showed positive expression for both *sycp3* and RNA polymerase II; round spermatids (SPD-r) and spermatozoa (SP) shared similar morphology under DAPI staining but differed in RNA polymerase II staining, with round spermatids being positive and spermatozoa negative. Between 50 and 100 cells were counted for each flow-sorted population, and only populations with an enrichment above 70% were considered for WGBS experiments.

## WGBS sample collection and library construction

Sorted cell populations were dissolved in homogenisation buffer (20 mM Tris pH 8.0, 100 mM NaCl, 15 mM EDTA, 1% SDS, 0.5 mg/ml Proteinase K) at 55 °C. Two Phenol/Chlorophorm/Isoamylalcohol (25:24:1, PCI) extractions were performed. DNA was precipitated using 1/5 volume of 4 M NH4Ac and 2.5 volumes of ice-cold absolute ethanol, and 1 μL of linear acrylamide. Samples were incubated overnight at −20 °C. The DNA was pelleted, resuspended in nuclease-free water and spiked with 0.1% λ DNA (Promega). Bisulfite converted libraries were generated using the Pico Methyl-Seq Library Prep Kit (Zymo Research, Cat. D5456) following the manufacturer's instructions. Libraries were sequenced on the Illumina NovaSeq X Plus Series platform (150 PE).

## WGBS analysis

Files were trimmed using Trimmomatic (v0.39) (HEADCROP:5 ILLUMINACLIP:TruSeq3-PE-2.fa:2:30:10 LEADING:3 TRAILING:3 SLIDINGWINDOW:4:15 MINLEN:50) (Bolger et al, 2014) and mapped using Bismark (v0.22.3) (--non_directional --local -X 2000) (Krueger and Andrews, 2011). Methylation was called using MethylDackel (v0.6.1) (--mergeContext --minOppositeDepth 10 --maxVariantFrac 0.5). DMRs were detected using DSS (v2.54.0) (delta=0.2, p.threshold=0.05, minlen=100, minCG=10, dis.-merge=100) (Feng et al, 2014).

## Data availability

The datasets produced in this study are available in the following databases: scATAC-seq data: Gene Expression Omnibus GSE283803. scRNA-seq data: Gene Expression Omnibus GSE283804. DNA methylation data: ArrayExpress E-MTAB-14873. The source data of this paper are collected in the following database record: biostudies:S-SCDT-10_1038-S44320-025-00157-7.

## Peer review information

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

## Acknowledgements

This work was supported by the Australian Research Council (ARC) Discovery Project DP190103852 to OB and by the Ministerio de Ciencia, Innovación y Universidades/Agencia Estatal de Investigación (MICIU/AEI, 10.13039/501100011033) through projects PID2021-128358NA-I00 to OB (co-funded by ERDF/EU), PID2022-141288NB-I00 to JJT (co-funded by ERDF/EU), PID2020-113647GA-I00 to MA-C, and PID2020-112557GB-I00 to AR-H; and by the Unidad de Excelencia María de Maeztu (CEX2020-001088-M, MICIU/AEI/10.13039/501100011033) to OB. OB further acknowledges a Ramón y Cajal fellowship (RYC2020-028685-I; MICIU/AEI/10.13039/501100011033 and the European Social Fund—"ESF Investing in your future"). AR-H also

acknowledges support from the Agència de Gestió d'Ajuts Universitaris i de Recerca (AGAUR, 2021SGR00122) and the Catalan Institution for Research and Advanced Studies (ICREA). GP was supported by an FPI predoctoral fellowship (PRE-C-2021-0083; MICIU/AEI/10.13039/501100011033 and ESF +). The authors thank the Garvan Genomics Platform for scATAC-seq and scRNA-seq library preparation, Nerea Roher for technical assistance, and members of the OB and MA-C laboratories for critical reading of the manuscript.

## Author contributions

**Ana María Burgos-Ruiz**: Data curation; Formal analysis; Investigation; Visualisation; Writing—review and editing. **Fan-Suo Geng**: Resources; Investigation; Writing—review and editing. **Gala Pujol**: Formal analysis; Investigation; Visualisation; Writing—review and editing. **Estefanía Sanabria**: Investigation; Writing—review and editing. **Thirsa Brethouwer**: Data curation; Formal analysis; Investigation; Writing—review and editing. **María Almuedo-Castillo**: Formal analysis; Investigation; Visualisation; Writing—review and editing. **Aurora Ruiz-Herrera**: Formal analysis; Investigation; Visualisation; Writing—review and editing. **Juan J Tena**: Conceptualisation; Data curation; Formal analysis; Supervision; Investigation; Visualisation; Writing—original draft; Writing—review and editing. **Ozren Bogdanovic**: Conceptualisation; Data curation; Formal analysis; Supervision; Funding acquisition; Investigation; Visualisation; Writing—original draft; Project administration; Writing—review and editing.

Source data underlying figure panels in this paper may have individual authorship assigned. Where available, figure panel/source data authorship is listed in the following database record: biostudies:S-SCDT-10_1038-S44320-025-00157-7.

## Disclosure and competing interests statement

The authors declare no competing interests.

# Expanded View Figures

**Figure EV1.  scRNA-seq quality control, marker genes and GO enrichments.**

(A) Quality control metrics for scRNA-seq replicates showing the relationship between counts, detected gene numbers, and mitochondrial DNA percentage. (B) UMAP (uniform manifold approximation and projection) plots of the zebrafish testis tissue denoted by Seurat clusters before filtering. (C) Distribution of average counts, and average gene numbers across the spermatogenic lineage. (D) Expression UMAP for canonical marker genes for SPG (*ddx4*), SPG-Aun (*eno3*, *e2f5* and *ripply2*), SPG-Ad(*hist1h2a6*), SPT-e (*edrf1*), SPC (*sycp3*) and proliferation (*pcna*). (E) Dot plot of the top 5 GO biological processes enriched using marker genes for each cell type. The x-axis shows adjusted *P* values (FDR), and dot size reflects the ratio of intersected to total terms. Enrichment *P* values were computed with Fisher's exact one-tailed test (cumulative hypergeometric).

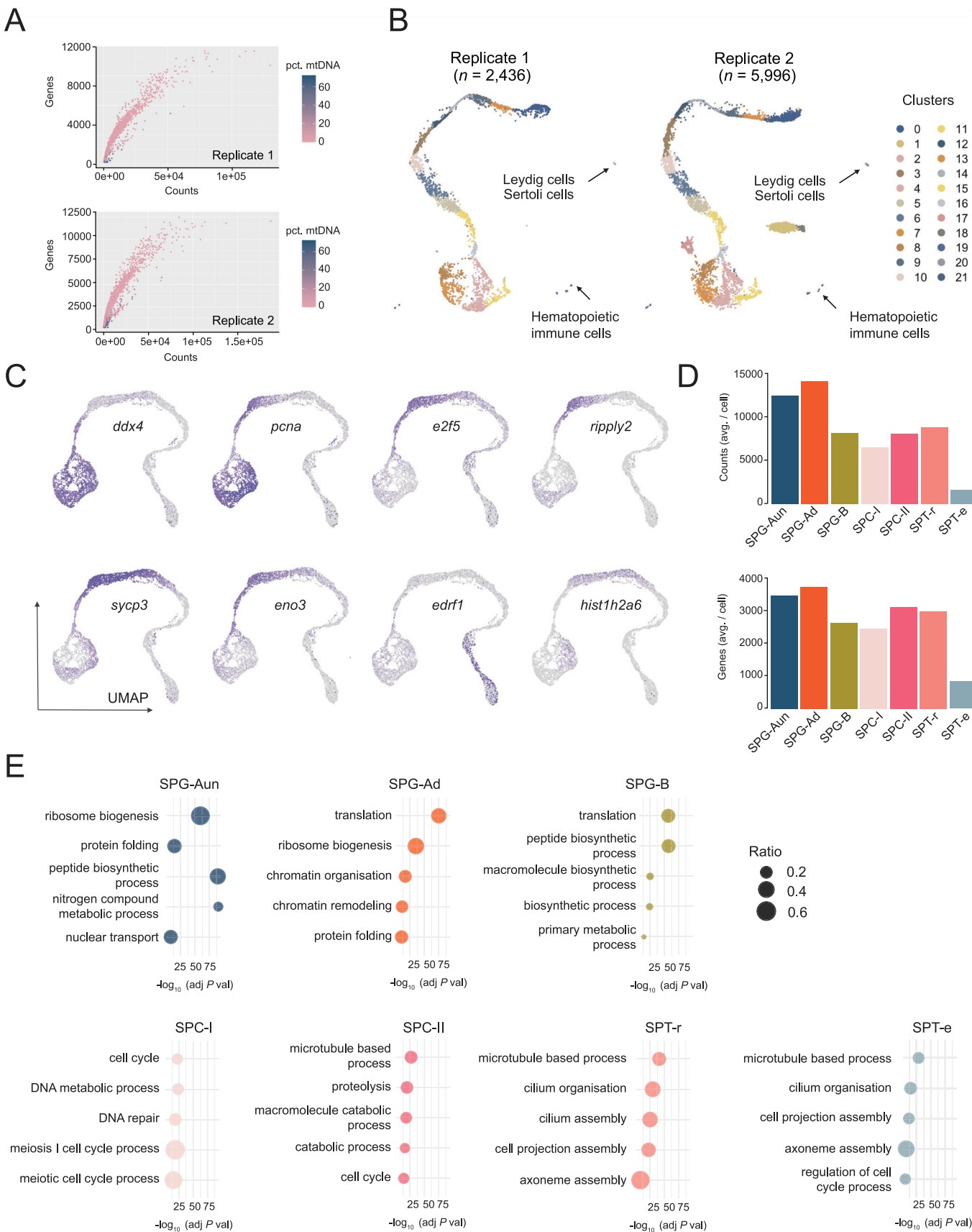

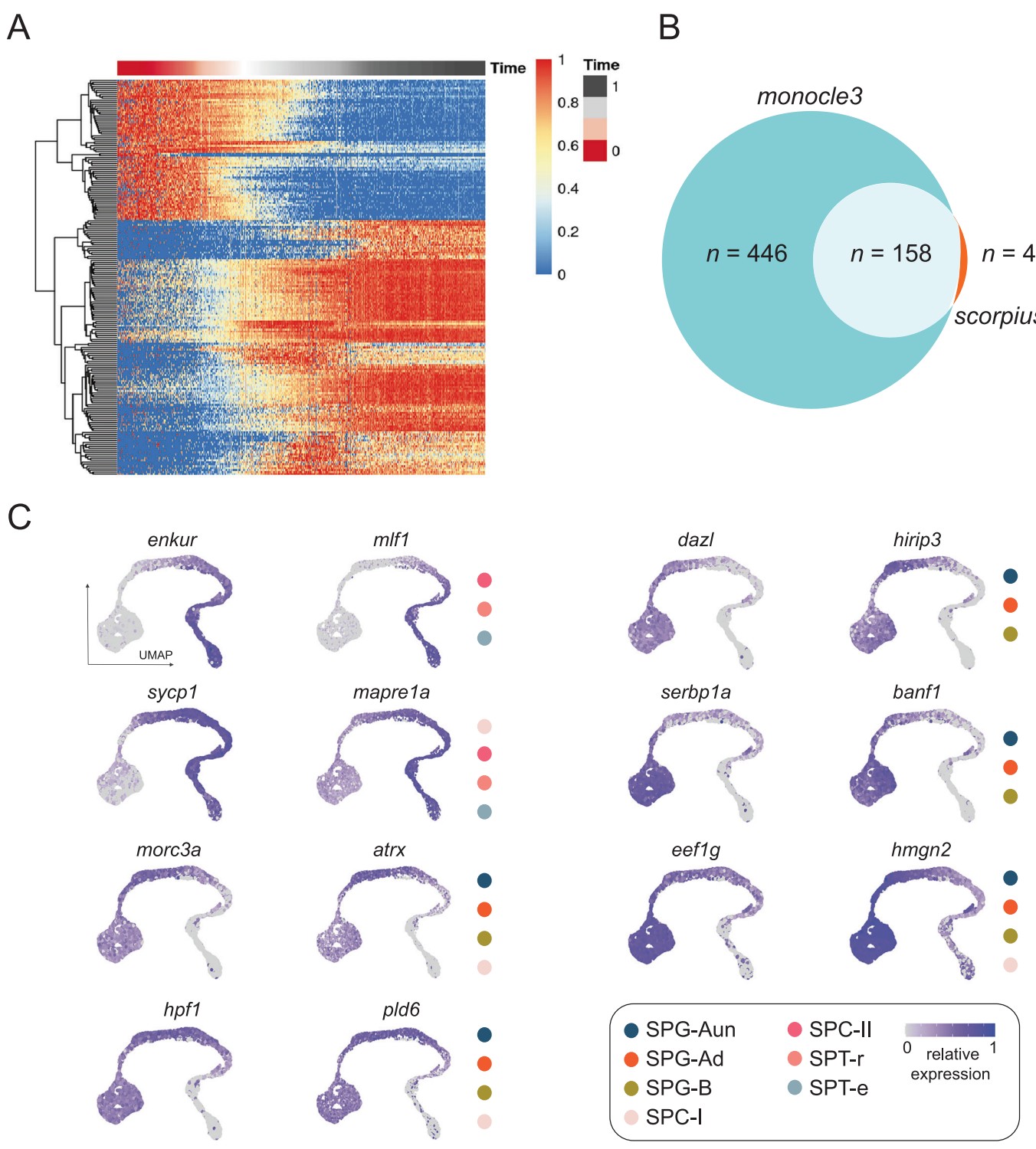

**Figure EV2.   Trajectory analysis and expression profiles of spermatogenesis driver genes.**

(A) Heatmap of drive genes across the inferred trajectory of spermatogenesis. Samples ranked by pseudotime trajectory inferred by SCORPIUS. Rows represent genes grouped by hierarchical clustering with distinct expression patterns. (B) Comparison of gene drivers for spermatogenesis calculated using Monocle3 and SCORPIUS software. (C) UMAP examples for gene drivers obtained from the trajectory analysis identified by both Monocle3 and SCORPIUS.

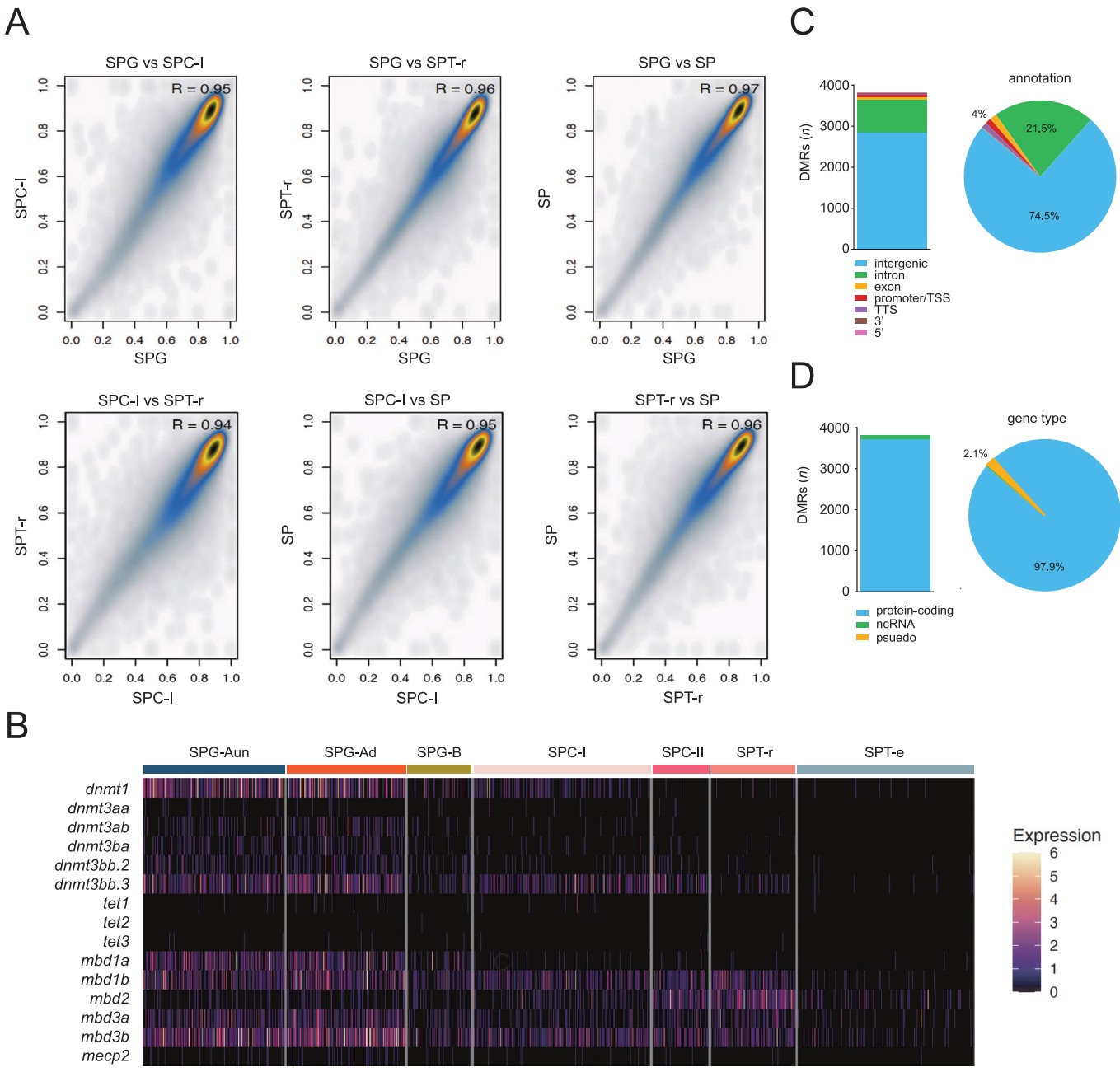

**Figure EV3.   DNA methylomes and genomic localisation of Differentially Methylated Regions (DMRs).**

(A) Correlation of CpG methylation levels in non-overlapping 10 Kb genomic bins in isolated germ cell populations (SPG: spermatogonia, SPC-I: spermatocytes I, SPT-r: round spermatids, and mature sperm: SP). *R* value represents Pearson Correlation coefficient. SPG vs SPC-I, SPC-I vs SPT-r, and SPT-r vs SP comparisons are also shown in Fig. 3E. (B) Single-cell RNA-seq expression of DNA methylation, deposition removal, and interpretation machinery during zebrafish spermatogenesis. Genes include DNA methyltransferases, TET dioxygenases, and methyl-CpG-binding domain proteins. Expression is shown across major germ cell populations, ordered by developmental progression from undifferentiated spermatogonia to mature spermatids. (C) Genomic localisation of DMRs (n = 3879) represented as absolute values (left panel) and percentages (pie chart, right panel) identified between four germ cell populations. (D) Gene type enrichment of the identified DMRs.

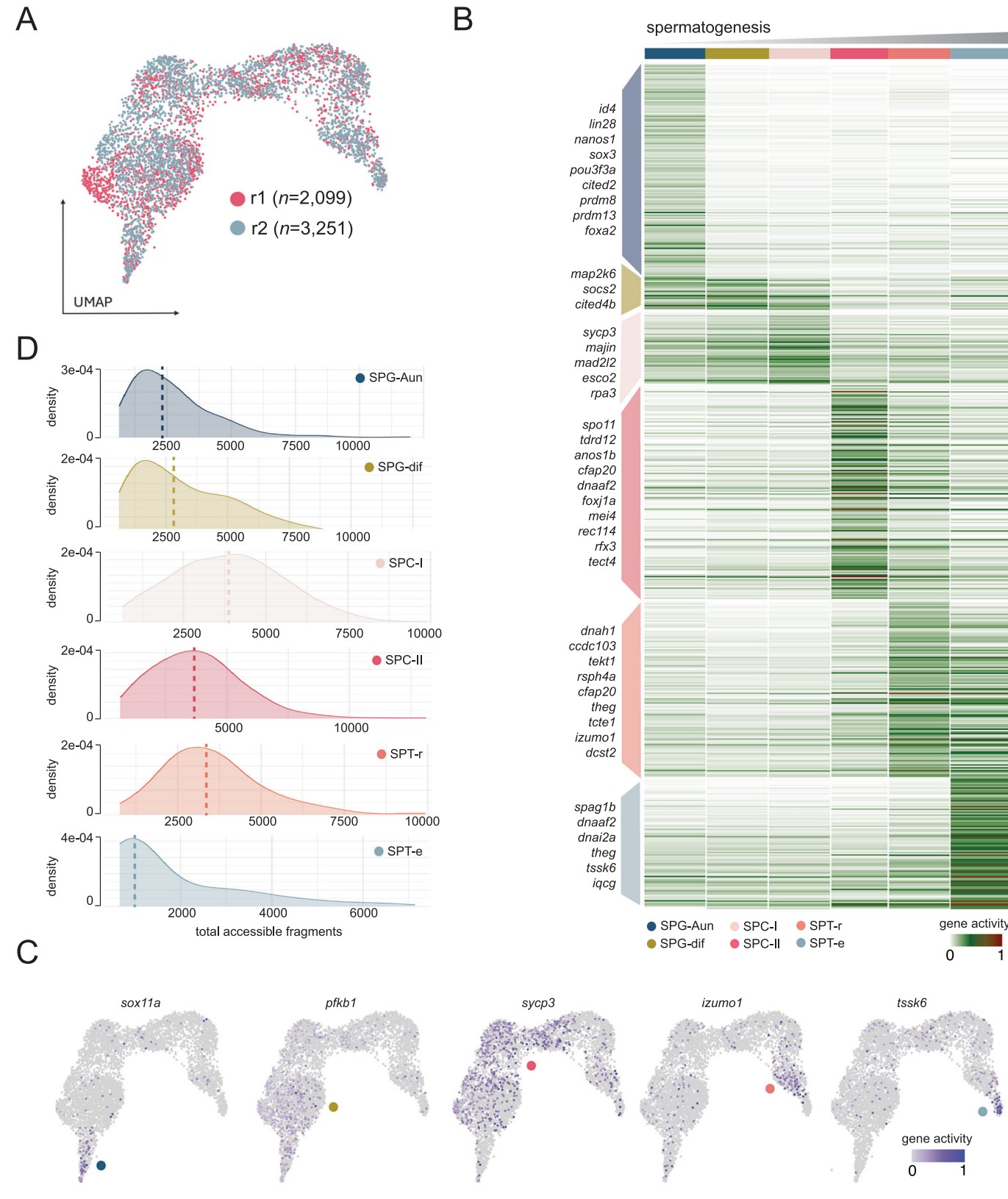

◀ **Figure EV4.   Single cell chromatin accessibility (scATAC-seq) dynamics during zebrafish spermatogenesis.**

(**A**) UMAP (uniform manifold approximation and projection) of biological replicates of scATAC-seq data. Cells are coloured by replicate label. (**B**) Sorted heatmap of averaged gene expression profiles grouped by cell type (SPG-Aun, SPG-dif, SPC-I, SPC-II, SPT-r, SPT-e). Top marker genes for each cluster are indicated on the left-hand side of the heatmap. (**C**) UMAP plots showing gene activity of marker gene examples associated with spermatogenesis: *sox11* - undifferentiated spermatogonia; *pfkb1* - differentiated spermatogonia; *sycp3* - spermatocytes; *izumo1* - round spermatids; *tssk6* - elongated spermatids. Gene activity is defined as the normalised sum of fragments spanning the gene body and extending 2 kb upstream of the transcription start site. (**D**) Distribution of the number of accessible genome fragments in each cell type (SPG-Aun, SPG-dif, SPC-I, SPC-II, SPT-r, SPT-e). The median is indicated by the dashed line.

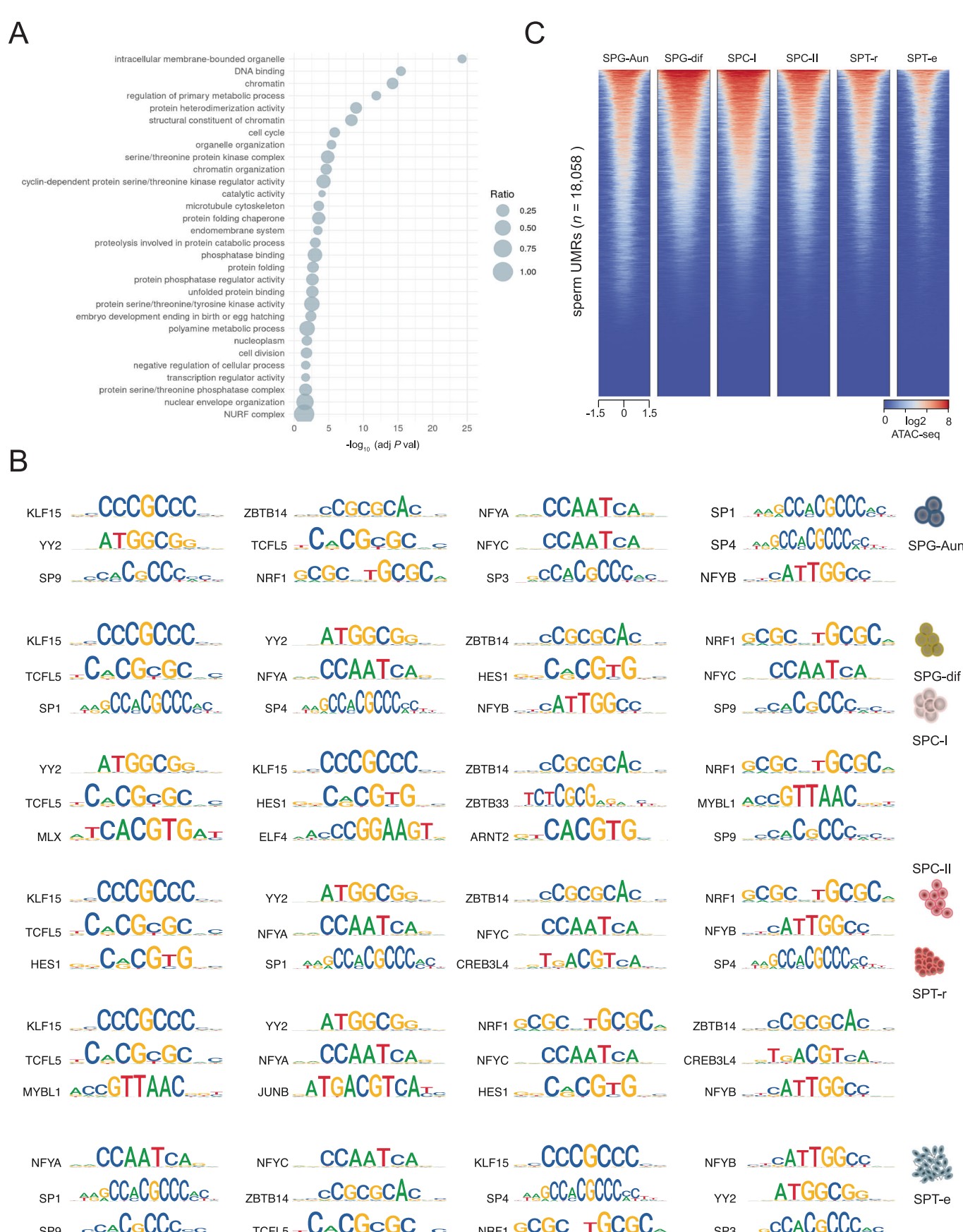

**Figure EV5.   Chromatin state of elongated spermatids (SPT-e).**

(A) Dot plot of top biological processes associated with genes linked to open chromatin peaks in the SPT-e population. The x-axis shows adjusted *P* values (FDR), and dot size reflects the ratio of intersected to total terms. Enrichment *P* values were computed with Fisher's exact one-tailed test (cumulative hypergeometric). (B) Top 12 enriched motifs found in open chromatin peaks in each cell population. (C) ATAC-seq signal (log2) from all spermatogenesis cell types, plotted over unmethylated regions (UMRs) identified in mature sperm.

