## [Peer Review File · Molecular Systems Biology]

A single-cell multiomics roadmap of zebrafish spermatogenesis reveals regulatory principles of male germline formation

Ana Burgos-Ruiz, Fan-Suo Geng, Gala Pujol, Estefanía Sanabria-Reinoso, Thirsa Brethouwer, Maria Almuedo-Castillo, Aurora Ruiz-Herrera, Juan Tena, and Ozren Bogdanovic

Corresponding author(s): Ozren Bogdanovic (o.bogdanovic@csic.es) , Juan Tena (juan.tena@csic.es)

Review Timeline:

Submission Date:	1st Apr 25
Editorial Decision:	20th May 25
Revision Received:	5th Aug 25
Editorial Decision:	12th Sep 25
Revision Received:	13th Sep 25
Accepted:	22nd Sep 25

Editor: Jingyi Hou

Transaction Report:

20th May 2025

Manuscript Number: MSB-2025-13014-T

Title: A single-cell multiomics roadmap of zebrafish spermatogenesis reveals regulatory principles of male germline formation

Author: Ana Burgos-Ruiz

Fan-Suo Geng

Gala Pujol

Estefanía Sanabria-Reinoso

Thirsa Brethouwer

María Almuedo-Castillo

Aurora Ruiz-Herrera

Juan Tena

Ozren Bogdanovic

Dear Dr. Bogdanovic,

Thank you for submitting your work to Molecular Systems Biology. I apologize for the somewhat slow process, which was due to delays in obtaining the reviewer reports. We have now heard back from the three reviewers who agreed to evaluate your manuscript. As you will see from the reports below, the reviewers think the study and the provided datasets interesting. However, they raised a series of concerns, which we would ask you to address in a major revision.

I think the reviewers' recommendations are clear, so it is unnecessary to reiterate the points listed below. In particular, Reviewer #2's Major Point #1 should be carefully addressed, either by tempering the conclusions related to evolution or by conducting additional evolutionary analyses to more robustly support them.

All other issues raised by the reviewers need to be satisfactorily addressed as well. As you may already know, our editorial policy allows in principle a single round of major revision, so it is essential to provide responses to the reviewers' comments that are as complete as possible. Please feel free to contact me in case you would like to discuss in further detail any of the issues raised by the reviewers.

On a more editorial level, we would ask you to address the following issues:

- Please provide a .docx formatted version of the manuscript text (including legends for main figures, EV figures and tables). Please make sure that the changes are highlighted to be clearly visible.
- Please provide individual production quality figure files as .eps, .tif, .jpg (one file per figure).
- Please provide a .docx formatted letter INCLUDING the reviewers' reports and your detailed point-by-point responses to their comments. As part of the EMBO Press transparent editorial process, the point-by-point response is part of the Review Process File (RPF), which will be published alongside your paper.
- Please note that all corresponding authors are required to supply an ORCID ID for their name upon submission of a revised manuscript.
- We replaced Supplementary Information with Expanded View (EV) Figures and Tables that are collapsible/expandable online (see examples in <http://msb.embopress.org/content/11/6/812>). A maximum of 5 EV Figures can be typeset. EV Figures should be cited as 'Figure EV1, Figure EV2' etc... in the text and their respective legends should be included in the main text after the legends of regular figures.

Additional Tables/Datasets should be labeled and referred to as Table EV1, Dataset EV1, etc. Legends have to be provided in a separate tab in case of .xls files. Alternatively, the legend can be supplied as a separate text file (README) and zipped together with the Table/Dataset file.

For the figures and tables that you do NOT wish to display as Expanded View figures, they should be bundled together with their legends in a single PDF file called *Appendix*, which should start with a short Table of Content. Each legend should be below the corresponding Figure/Table in the Appendix. Appendix figures and tables should be referred to in the main text as: "Appendix Figure S1, Appendix Figure S2, Appendix Table S1" etc. See detailed instructions regarding expanded view here: <https://www.embopress.org/page/journal/17444292/authorguide#expandedview>.

- Before submitting your revision, primary datasets (and computer code, where appropriate) produced in this study need to be deposited in an appropriate public database (see <http://msb.embopress.org/authorguide> - dataavailability

<https://www.embopress.org/page/journal/17444292/authorguide#dataavailability>).

The accession numbers and database should be listed in a formal "Data Availability" section (placed after Materials & Method) that follows the model below (see also <https://www.embopress.org/page/journal/17444292/authorguide#dataavailability>). Please note that the Data Availability Section is restricted to new primary data that are part of this study.

Data availability

- RNA-Seq data: Gene Expression Omnibus GSE46843 (<https://www.ncbi.nlm.nih.gov/geo/query/acc.cgi?acc=GSE46843>)

- [data type]: [name of the resource] [accession number/identifier/doi] ([URL or identifiers.org/DATABASE:ACCESSION])

-At EMBO Press we ask authors to provide source data for the main figures. Our source data coordinator will contact you to discuss which figure panels we would need source data for and will also provide you with helpful tips on how to upload and organize the files.

- Our journal encourages inclusion of *data citations in the reference list* to directly cite datasets that were re-used and obtained from public databases. Data citations in the article text are distinct from normal bibliographical citations and should directly link to the database records from which the data can be accessed. In the main text, data citations are formatted as follows: "Data ref: Smith et al, 2001". In the Reference list, data citations must be labeled with "[DATASET]". A data reference must provide the database name, accession number/identifiers and a resolvable link to the landing page from which the data can be accessed at the end of the reference. Further instructions are available at .

- We updated our journal's competing interests policy in January 2022 and request authors to consider both actual and perceived competing interests. Please review the policy <https://www.embopress.org/competing-interests> and update your competing interests if necessary.

Please use the heading "Disclosure statement and competing interests".

- All Materials and Methods need to be described in the main text using our 'Structured Methods' format. According to this format, the Methods section includes a Reagents and Tools Table (listing key reagents, experimental models, software and relevant equipment and including their sources and relevant identifiers) followed by a Methods and Protocols section describing the methods, ideally using a step-by-step protocol format. The aim is to facilitate adoption of the methodologies across labs.

Please download and fill our Reagents and Tools Table template (.docx), which you can find in our author guidelines:

<https://www.embopress.org/page/journal/17444292/authorguide#structuredmethods>.

-Regarding data quantification:

Please ensure to specify the name of the statistical test used to generate error bars and P values, the number (n) of independent experiments (please specify technical or biological replicates) underlying each data point and the test used to calculate p-values in each figure legend. Discussion of statistical methodology can be reported in the materials and methods section, but figure legends should contain a basic description of n, P and the test applied.

Graphs must include a description of the bars and the error bars (s.d., s.e.m.).

- Please provide a "standfirst text" summarizing the study in one or two sentences (approximately 250 characters, including space), three to four "bullet points" highlighting the main findings and a "synopsis image" (550px width and 400-600 px height, PNG format) to highlight the paper on our homepage.

Here are a couple of examples:

<https://www.embopress.org/doi/10.15252/msb.20199356>

<https://www.embopress.org/doi/10.15252/msb.20209475>

<https://www.embopress.org/doi/10.15252/msb.209495>

When you resubmit your manuscript, please download our CHECKLIST (<https://www.embopress.org/pb-assets/embosite/EMBO%20Press%20Author%20Checklist-1642513524327.xlsx>) and include the completed form in your submission.

Please note that the Author Checklist will be published alongside the paper as part of the transparent process (<https://www.embopress.org/page/journal/17444292/authorguide#transparentprocess>).

If you feel you can satisfactorily deal with these points and those listed by the referees, you may wish to submit a revised version of your manuscript. Please attach a covering letter giving details of the way in which you have handled each of the points raised

by the referees. A revised manuscript will be once again subject to review and you probably understand that we can give you no guarantee at this stage that the eventual outcome will be favorable.

I look forward to receiving the revised manuscript soon.

Kind regards,
Jingyi

Jingyi Hou, PhD
Senior Editor
Molecular Systems Biology

We realize that it is difficult to revise to a specific deadline. In the interest of protecting the conceptual advance provided by the work, we recommend a revision within 3 months (18th Aug 2025). Please discuss the revision progress ahead of this time with the editor if you require more time to complete the revisions. Use the link below to submit your revision:

IMPORTANT: When you send your revision, we will require the following items:

1. the manuscript text in LaTeX, RTF or MS Word format
2. a letter with a detailed description of the changes made in response to the referees. Please specify clearly the exact places in the text (pages and paragraphs) where each change has been made in response to each specific comment given
3. three to four 'bullet points' highlighting the main findings of your study
4. a short 'blurb' text summarizing in two sentences the study (max. 250 characters)
5. a 'thumbnail image' (550px width and max 400px height, Illustrator, PowerPoint or jpeg format), which can be used as 'visual title' for the synopsis section of your paper.
6. Please include an author contributions statement after the Acknowledgements section (see <https://www.embopress.org/page/journal/17444292/authorguide>)
7. Please complete the CHECKLIST available at (<https://bit.ly/EMBOPressAuthorChecklist>). Please note that the Author Checklist will be published alongside the paper as part of the transparent process (<https://www.embopress.org/page/journal/17444292/authorguide#transparentprocess>).
8. When assembling figures, please refer to our figure preparation guideline in order to ensure proper formatting and readability in print as well as on screen:
<https://bit.ly/EMBOPressFigurePreparationGuideline>
See also figure legend guidelines: <https://www.embopress.org/page/journal/17444292/authorguide#figureformat>
9. Please note that corresponding authors are required to supply an ORCID ID for their name upon submission of a revised manuscript (EMBO Press signed a joint statement to encourage ORCID adoption). (<https://www.embopress.org/page/journal/17444292/authorguide#editorialprocess>)
Currently, our records indicate that the ORCID for your account is 0000-0001-5680-0056.

Please click the link below to modify this ORCID:
Link Not Available

11. Include a Reagents and Tools Table which can be downloaded from our author guidelines (<https://www.embopress.org/page/journal/17444292/authorguide#structuredmethods>)

*** PLEASE NOTE *** As part of the EMBO Press transparent editorial process initiative (see our Editorial at <https://dx.doi.org/10.1038/msb.2010.72>), Molecular Systems Biology publishes online a Review Process File with each accepted manuscripts. This file will be published in conjunction with your paper and will include the anonymous referee reports, your point-by-point response and all pertinent correspondence relating to the manuscript. If you do NOT want this File to be published, please inform the editorial office at contact@molsystbiol.org within 14 days upon receipt of the present letter.

Reviewer #1:

Ruiz et al. combine scRNA-seq, scATAC-seq and whole genome bisulfite sequencing to characterize the genomic and transcriptomic state of adult zebrafish sperm development in the testis. By combining these technologies, the authors were able to:

[1] map the trajectories of spermatogenesis based on transcriptome states (while identifying all the cell types that would be expected),

[2] assess the methylation status of male zebrafish germline development,

[3] examine chromatin accessibility across differentiation and

[4] correlate open chromatin/hypomethylated regions to placeholder nucleosome positioning to facilitate the next generation.

Overall, this work is a tour de force and provides a rich dataset to probe this unique developmental process that capitalizes on the constant differentiation status of the male zebrafish germline (from early to late differentiation states). The findings in this manuscript are consistent with mammalian findings, however they fill the gap in our knowledge of how this process functions in a system devoided of protamines such as zebrafish. Furthermore, this work contributes to a further understanding of how intergenerational epigenetic inheritance may function in zebrafish.

General Remarks

The paper was well written, clear and presented the data in a balanced manner. While other groups have performed similar experimental designs for mapping spermatogenesis with single-cell technology (e.g. Sposato et al. 2024, Development and Qian et al. 2022, Front Genet), this paper dives deep into multiple modalities to paint a much clearer picture. The technical advances were most clear when these modalities were correlated with each other (i.e. chromatin accessibility and methylation status). Many audiences will find these data to be important to analyze, from germline specialists to evolutionary biologists. Given these points, I strongly support publication of this manuscript.

Major Points

- Given the TF modules identified to be enriched in the open chromatin regions, are these TFs specifically expressed in the germline or are they somatically expressed as well? i.e. NFY TFs visualized by FISH in the germline. This would strengthen their argument as being potent drivers for this differentiation trajectory.

- Many cells were discarded from the analysis, such as 17392 to 5350 in the scATAC dataset. Are these the result of overloading the droplets, cell death during suspension prep, contaminating cells etc?

- Are there candidate methylase/demethylase enzymes that can be IDed in the single cell expression that may be driving the localized changes in the DMRs?

Minor Points

- While the manuscript in its presented form looks fantastic, it's more helpful to reviewers to have a document with line numbers for referencing specific regions. Something for future consideration!

- Figures were all beautifully displayed, well organized and clear. Some of the fluorescent signals in figure 2D were hard to see in the overlay and could be improved by changing the red-green channels to a more easily visible colour panel (green and magenta for example, with overlap being white)

- As aging was previously found to be a major determinant for spermatogenesis progression in zebrafish using similar approaches, it may be worthwhile to speculate more in the discussion how these additional modalities might shed light onto that process.

Reviewer #2:

In this study, Burgos-Ruiz et al. explore transcriptional and regulatory dynamics of zebrafish spermatogenesis by producing and analyzing a comprehensive single-cell multiomic resource (scRNA-seq and scATAC-seq) as well as DNA base-resolution DNA methylome (WGBS) of sorted germ cell populations from zebrafish testes. Notably, they identified major cell types involved in zebrafish spermatogenesis (mainly germ cells) along with associated marker genes, transcriptional activity, chromatin accessibility and DNA methylation.

In my opinion, this study is mainly descriptive and the advances lie in the dataset itself. Indeed, the generated dataset is highly relevant for the evo-devo scientific community as it provides unprecedented transcriptional and epigenetic data for spermatogenesis in an amniote species. However, I have some points I would like the authors to clarify/improve.

Major points:

1) Although this dataset is appropriate to better understand the evolution of spermatogenesis in vertebrates, the evolutionary conclusions drawn here are somehow overstated, in my opinion, as the evolutionary analyses remain superficial. Thus, I would recommend to either tune down the evolutionary conclusions all over the manuscript or perform cross-species analyses integrating amniote and anamniote species. For example, it is difficult to claim "Collectively, these observations underscore the

deeply conserved nature of global chromatin state changes across vertebrates and their importance in shaping male germ cell development" without a proper cross-species comparison.

2) It is very unfortunate that the dataset and analyses focus on germ cells only. However, some somatic cells are identified as shown in Figure S1. It would be relevant to provide metrics for those cell populations (number of cells, number of detected genes, marker genes...) and discuss why so few can be retrieved. Also, in replicate 2, a distinct cluster potentially corresponding to somatic cells is not discussed. What is this cluster?

3) Usable droplets are identified by using the CellRanger pipeline. It would be relevant to provide the CellRanger reports as well as figures associated with QC analyses to better assess the quality of the data. For example, elongated spermatids are almost entirely transcriptionally silent, how did the authors distinguish them from empty droplets?

4) It is not possible to access the data as it is protected by a password. I don't know whether I missed the information or if it has not been provided to me.

5) Three populations of spermatogonia have been identified. It would be relevant to perform analyses on spermatogonia dynamics along with SSC self-renewal and differentiation as the dataset seem appropriate to do so. Indeed, the level of conservation of these processes across vertebrates is of high interest for the evo-devo community.

Minor points:

1) The quality of figure S5A is very low as it is not possible to read the labels.

2) Why does the data come from transgenic zebrafish individuals?

3) Additional metrics such as the percentage of intronic reads along spermatogenesis would be interesting as it reflects active transcription.

Reviewer #3:

Ana María Burgos Ruíz and colleagues developed a comprehensive single-cell multi-omics resource that combines single-cell RNA sequencing (scRNA-seq) and single-cell chromatin accessibility (scATAC-seq) profiling in zebrafish spermatogenesis with base-resolution DNA methylome (WGBS) analysis of sorted germ cell populations. scRNA-seq annotated the major germ cell populations: undifferentiated spermatogonia A (SPG-Aun), differentiated spermatogonia A (SPG-Ad), spermatogonia B (SPG-B), primary spermatocytes (SPC-I), secondary spermatocytes (SPC-II), round spermatids (SPT-r), and elongated spermatids (SPT-e), and identified a lot of new driver genes.

Next, to gain insight into gene regulation during spermatogenesis, we performed DNA methylation (5mCG) and found thousands of localized and bidirectional 5mCG changes in potential gene regulatory regions that occur at the spermatocyte stage. To further study the gene regulation mechanisms operating during spermatogenesis, they analyzed scATAC-seq libraries and found local chromatin changes associated with the activity of spermatogenic marker genes, as well as a stepwise reprogramming of global chromatin structure that leads to chromatin compaction and transcriptional shutdown.

Finally, they investigated the chromatin accessibility profile of elongated spermatids using scATAC-seq and found that late SPT-e ATAC-seq signals were indeed strongly consistent with placeholder chromatin, which was identified as a major driver of maternal-to-paternal chromatin remodeling before ZGA. These findings are extremely useful for understanding the changes in chromosome structure accompanying changes in gene expression patterns during spermatogenesis, and provide valuable insight for understanding the chromosome structure of spermatozoa that transmit genome to the next generation in non-mammals.

The results are comprehensive and have been tested in biological duplicate. Therefore, they are highly reliable, but I concern the number of SPG-Aun cells (Fig. 1D). In general, it is known that the number of germline stem cells, as well as undifferentiated germ cells, is overwhelmingly smaller than that of differentiated germ cells. In fact, Leal et al. (2009) reported that the number of undifferentiated germ cells was also small in histological observations. Authors should have an explanation for this difference. It is also confusing that the notation of SPG is different for scRNA-seq, DNA methylome (WGBS) analysis, and scATAC-seq. The notation should be consistent throughout the paper, and if that is difficult, each should be clearly defined.

Minor point

From the first line from the bottom of the left paragraph to the fourth line from the top of the right paragraph on page 4, please clearly indicate which populations are SPG populations and SPC cells.

The image quality of the supplemental figures is so low that Fig. S5A is almost unreadable.

Point by point response to Reviewers' comments - MSB-2025-13014-T**Reviewer #1:**

Ruíz et al. combine scRNA-seq, scATAC-seq and whole genome bisulfite sequencing to characterize the genomic and transcriptomic state of adult zebrafish sperm development in the testis. By combining these technologies, the authors were able to:

- [1] map the trajectories of spermatogenesis based on transcriptome states (while identifying all the cell types that would be expected),
- [2] assess the methylation status of male zebrafish germline development,
- [3] examine chromatin accessibility across differentiation and
- [4] correlate open chromatin/hypomethylated regions to placeholder nucleosome positioning to facilitate the next generation.

Overall, this work is a tour de force and provides a rich dataset to probe this unique developmental process that capitalizes on the constant differentiation status of the male zebrafish germline (from early to late differentiation states). The findings in this manuscript are consistent with mammalian findings, however they fill the gap in our knowledge of how this process functions in a system devoided of protamines such as zebrafish. Furthermore, this work contributes to a further understanding of how intergenerational epigenetic inheritance may function in zebrafish.

General Remarks

The paper was well written, clear and presented the data in a balanced manner. While other groups have performed similar experimental designs for mapping spermatogenesis with single-cell technology (e.g. Sposato et al. 2024, *Development* and Qian et al. 2022, *Front Genet*), this paper dives deep into multiple modalities to paint a much clearer picture. The technical advances were most clear when these modalities were correlated with each other (i.e. chromatin accessibility and methylation status). Many audiences will find these data to be important to analyze, from germline specialists to evolutionary biologists. Given these points, I strongly support publication of this manuscript.

ANSWER: We thank the Reviewer for their supportive comments and helpful suggestions, which have improved the quality of our manuscript.

Major Points

Given the TF modules identified to be enriched in the open chromatin regions, are these TFs specifically expressed in the germline or are they somatically expressed as well? i.e. NFY TFs visualized by FISH in the germline. This would strengthen their argument as being potent drivers for this differentiation trajectory.

ANSWER: The CpG island-binding proteins we identified as enriched in elongated spermatids (based on transcription factor motif enrichment in ATAC-seq peaks; Figure 5A) are broadly expressed chromatin-associated factors, with no apparent germline specificity. We believe that these proteins play key roles in the general maintenance of open chromatin structure throughout the vertebrate life cycle and as speculated, may even contribute to the transmission of chromatin states across generations. Notably, this set of proteins does not overlap with the spermatogenesis driver genes shown in Figure 2. To further illustrate this distinction, we have now added **Appendix Figure S7**, which shows the expression profiles of these proteins across embryonic, adult somatic, and germline tissues. The changes are reflected in the following sentence:

P7: *"Importantly, many of these broadly expressed factors (Appendix Figure S7) (Tapial et al, 2017), as well as other key components of CGI chromatin, were strongly expressed in mature sperm and zebrafish testis tissues..."*

Many cells were discarded from the analysis, such as 17392 to 5350 in the scATAC dataset. Are these the result of overloading the droplets, cell death during suspension prep, contaminating cells etc?

ANSWER: We thank the Reviewer for raising this point. A summary of the breakdown of cell loss is now provided as **Appendix Table S2**. By applying a minimum threshold of 500 unique fragments per barcode, we removed low-quality barcodes unlikely to correspond to real cells (1628 cells in Sample 1 and 441 in Sample 2), thereby excluding empty droplets and background DNA. To avoid inclusion of overloaded droplets or potential multiplets, we set an upper limit of 10,000 fragments, which excluded 164 cells in Sample 1 and 286 in Sample 2. We further required that at least 40% of reads fall within called peaks to enrich for high-confidence chromatin accessibility profiles, removing 346 and 615 cells from Samples 1 and 2, respectively. To eliminate profiles dominated by nucleosomal fragments, often indicative of dying or compromised cells, we excluded cells with a nucleosome signal ≥ 1 , which removed an additional 1636 cells in Sample 1 and 518 in Sample 2. A transcription start site (TSS) enrichment score above 5 was used to retain cells with strong promoter-proximal signal and low background, excluding 1265 and 2233 cells from Samples 1 and 2, respectively. Finally, we filtered out cells flagged as potential doublets by scDbtFinder, resulting in the removal of 273 cells in Sample 1 and 554 in Sample 2. These quality-control steps were chosen to prioritize data reliability over cell number, particularly given that we had two biological replicates to buffer against cell loss. While stringent, we believe that these filters were essential to ensure that downstream analyses were based on high-quality, reproducible chromatin accessibility profiles.

P6: *"For scATAC-seq, we sequenced a total of 17,392 single cells from two biological replicates, and after filtering for doublets and low-quality cells (Appendix Table 2), we obtained 2,099 cells for replicate one and 3,251 cells for replicate two."*

Are there candidate methylase/demethylase enzymes that can be IDed in the single cell expression that may be driving the localized changes the DMRs?

ANSWER: This is a valuable suggestion. In response, we have now analyzed the expression patterns of DNMTs, TETs, and MBDs to gain insight into the dynamics and interpretation of DNA methylation during spermatogenesis. These data are now presented in **Extended View Figure 3B**, and the following paragraph has been added to the manuscript:

P5: *"The identified DMRs likely arise from a combination of active de novo methylation, driven mainly by high expression of dnmt3bb.2, and passive demethylation, potentially due to reduced dnmt1 expression during these stages (Fig. EV3B). The absence of tet1, tet2, and tet3 expression in spermatocytes supports the view that active DNA demethylation is unlikely to contribute significantly to methylation dynamics during this stage of spermatogenesis. Instead, changes in 5mCG are more likely driven by a combination of de novo methylation and passive mechanisms. In addition, the differential expression of methyl-CpG binding domain (MBD) proteins suggests stage-specific interpretation of methylation marks; mbd1a is predominantly expressed in spermatogonia (SPG-Aun, SPG-Ad and SPG-B), whereas mbd2 is selectively expressed at later stages, including SPC-II and SPT-r."*

Minor Points

While the manuscript in its presented looks fantastic, it's more helpful to reviewers to have a document with line numbers for referencing specific regions. Something for future consideration!

ANSWER: We apologize for this oversight. In our experience, page numbering is typically added during the PDF conversion at the time of submission. It is unclear why this did not occur in the present case. We appreciate the Reviewer for bringing this to our attention and will take extra care with future submissions.

Figures were all beautifully displayed, well organized and clear. Some of the fluorescent signals in figure 2D were hard to see in the overlay and could be improved by changing the red-green channels to a more easily visible colour panel (green and magenta for example, with overlap being white).

ANSWER: We have now changed the channels to green and magenta with white overlap, as per Reviewer's suggestion.

As aging was previously found to be a major determinant for spermatogenesis progression in zebrafish using similar approaches, it may be worthwhile to speculate more in the discussion how these additional modalities might shed light onto that process.

ANSWER: We have now added a paragraph to the Discussion to highlight how future studies using scATAC-seq and WGBS could further elucidate the epigenetic mechanisms underlying age-associated defects in zebrafish spermatogenesis.

P9: *“Future studies using scATAC-seq and WGBS could also elucidate whether the age-associated transcriptional changes observed in zebrafish spermatogonia (Sposato et al., 2024), such as repression of *pivill1*, *e2f5*, and *ube2* genes and aberrant activation of *pou5f3*, *nanog*, and *zp3.2*, are driven by alterations in chromatin accessibility or DNA methylation at key regulatory loci. Such approaches could clarify whether the observed lineage infidelity and impaired differentiation arise from errors in epigenetic reprogramming or stochastic loss of regulatory fidelity.”*

Reviewer #2:

In this study, Burgos-Ruiz et al. explore transcriptional and regulatory dynamics of zebrafish spermatogenesis by producing and analyzing a comprehensive single-cell multiomic resource (scRNA-seq and scATAC-seq) as well as DNA base-resolution DNA methylome (WGBS) of sorted germ cell populations from zebrafish testes. Notably, they identified major cell types involved in zebrafish spermatogenesis (mainly germ cells) along with associated marker genes, transcriptional activity, chromatin accessibility and DNA methylation.

In my opinion, this study is mainly descriptive, and the advances lie in the dataset itself. Indeed, the generated dataset is highly relevant for the evo-devo scientific community as it provides unprecedented transcriptional and epigenetic data for spermatogenesis in an amniote species. However, I have some points I would like the authors to clarify/improve.

ANSWER: We appreciate the Reviewer's positive feedback and valuable suggestions, which have contributed to strengthening our manuscript.

Major points:

Although this dataset is appropriate to better understand the evolution of spermatogenesis in vertebrates, the evolutionary conclusions drawn here are somehow overstated, in my opinion, as the evolutionary analyses remain superficial. Thus, I would recommend to either tune down the evolutionary conclusions all over the manuscript or perform cross-species analyses integrating amniote and anamniote species. For example, it is difficult to claim "Collectively, these observations underscore the deeply conserved nature of global chromatin state changes across vertebrates and their importance in shaping male germ cell development" without a proper cross-species comparison.

ANSWER: We have now removed this sentence from the manuscript as well as any notion to evolutionary analyses. We have also rephrased a couple of sentences to better capture this point.

P1: *“Overall, this high-resolution atlas of zebrafish spermatogenesis provides a valuable resource for studying vertebrate germ cell development, evolution, and epigenetic inheritance.”* has now been replaced with:

“In summary, this high-resolution atlas of zebrafish spermatogenesis provides a valuable resource for studying vertebrate germ cell development and epigenetic inheritance, while offering a robust framework for comparative analyses across diverse models of germ cell biology.”

P9: *“Overall, we present a significant community resource that advances understanding of vertebrate spermatogenesis, epigenetic inheritance, and the evolutionary conservation of gene regulatory processes associated with germ cell development.”* has now been changed to:

“Our study thus provides a valuable first step toward understanding gene regulatory dynamics during spermatogenesis in anamniotes and offers a framework for addressing fundamental questions in the regulation of this process.”

2) It is very unfortunate that the dataset and analyses focus on germ cells only. However, some somatic cells are identified as shown in Figure S1. It would be relevant to provide metrics for those cell populations (number of cells, number of detected genes, marker genes...) and discuss why so few can be retrieved.

ANSWER: We thank the Reviewer for raising this point. We have now conducted a comprehensive analysis of somatic cell types present in the zebrafish testis. Regarding somatic cell abundance, our findings are consistent with previous reports. Qian et al. identified somatic cells as comprising approximately 1.3% of the total testicular cell population. Sposato et al. reported that germ cells made up more than ~93% of their dataset, although exact numbers were not provided. This suggests that somatic cells, excluding potential contamination, likely account for 1-5% of total cells in the study by Sposato et al. In our data, now detailed in **Dataset EV1** and **Appendix Figures S2–S4**, we identified a total of 125 somatic cells out of 6,880 cells, corresponding to ~1.8%, thus aligning with previously published estimates. The lower proportion of somatic cells compared to mammalian studies (e.g., Green et al., 2018; Shami et al., 2020) may reflect species-specific differences, variation in testis developmental stage, or technical biases related to dissociation sensitivity. In particular, the large size and irregular morphology of some somatic cell types may have reduced their capture efficiency during 10x Genomics scRNA-seq, a limitation also noted by Sposato et al.

P3: *“Most cells belonged to germ cell populations, however, small populations of somatic cells (n=125) (Appendix Figs. S2A and S2B), comprising Leydig (expressed markers: *insl3*, *cyp11a1*, *cyp17*, *star*, *hsd3bl*, and others) (Tremblay, 2015) (Appendix Fig. S2C and Dataset EV1), Sertoli (expressed markers: *krt18a.1*, *aldh1a2*, *fxyd6*, *olfml3*, *cxcl12* and others) (De Gendt et al, 2014; Gilbert et al, 2009;*

*Qian et al., 2022; Raverdeau et al., 2012) (Appendix Fig. S3 and Dataset EV1), and hematopoietic immune cells (expressed markers: *rac2*, *coro1a*, *grap2* and others) (Deng et al., 2011; Li et al., 2012; Ma et al., 2001) (Appendix Fig. S4 and Dataset EV1), were also identified. Interestingly, a subpopulation of Sertoli cells displayed specific expression of genes such as *gstt1a*, *gpx3*, *uraha*, *aqp3*, *gsdf* and others, as revealed by UMAP feature plots (Appendix Fig. S3B). These genes are associated with oxidative stress response, solute transport, and signalling functions, suggesting that this subset may represent a metabolically specialized or stress-responsive state. Alternatively, the divergence might reflect spatial heterogeneity within the testis, or a temporary functional program related to germ cell interaction. Overall, the identified proportion of somatic cells (1.8%) is in line with previous zebrafish studies (Qian et al., 2022; Sposato et al., 2024); however, we acknowledge that this is likely an underrepresentation due to technical limitations associated with cell size and tissue dissociation, as noted previously (Sposato et al., 2024).”*

Also, in replicate 2, a distinct cluster potentially corresponding to somatic cells is not discussed. What is this cluster?

ANSWER: After careful examination, we decided to exclude this cluster from our downstream analyses for several reasons. First, this cluster was only observed in a single replicate and did not appear in any of the other samples, suggesting that it may not represent a consistent or biologically relevant population. Second, analysis of its marker genes revealed no strong or specific expression signature corresponding to any well-characterized testicular or somatic cell type. Additionally, we observed increased expression of apoptosis-related genes in this cluster, including *casp3.1*, *casp8*, *bax*, *tp53*, *chac1*, and *cdkn1a*, suggesting enrichment in dying or stressed cells. Taken together, these findings led us to conclude that the cluster was unlikely to represent a genuine or interpretable cell population, and we therefore excluded it from further analysis to ensure the robustness and biological relevance of our results.

Usable droplets are identified by using the Cell Ranger pipeline. It would be relevant to provide the Cell Ranger reports as well as figures associated with QC analyses to better assess the quality of the data.

ANSWER: As requested, we now include the summary of the Cell Ranger pipeline as **Extended View Table EV1**, which provides key metrics for each sample. While the parameters used for QC filtering were already described in the main text, we agree that including graphical representations provides a clearer assessment of data quality. We have also compiled a set of QC plots illustrating the distributions of key metrics for scRNAseq and scATACseq samples as **Appendix Fig. S1**.

For example, elongated spermatids are almost entirely transcriptionally silent, how did the authors distinguish them from empty droplets?

ANSWER: While it is true that elongated spermatids are largely transcriptionally silent, several lines of evidence support the validity of this population in our dataset. First, we consistently identified the same population of elongated spermatids across two independently processed biological samples, arguing against the possibility that these cells represent artifacts or ambient RNA contamination. Second, all cells included in the analysis passed stringent quality control criteria, including the exclusion of cells with fewer than 200 detected genes, thereby minimizing the likelihood of retaining empty droplets. Although transcriptionally quiescent, elongated spermatids still express key genes required for flagellar motility. Cells annotated as SPT-e express canonical markers of elongated spermatids (**Dataset EV2**), including *tssk6*, *spag6*, *spag1b*, *rfx2*, *rfx3*, and *theg*, supporting the accuracy of their annotation. Importantly, scATAC-seq data further corroborate these findings, revealing accessible chromatin at

several of these key loci, such as *spag1b*, *tssk6*, and *theg* (**Dataset EV6 and Fig. EV4**). Taken together, these observations provide strong evidence that the identified population corresponds to *bona fide* elongated spermatids rather than technical artifacts.

It is not possible to access the data as it is protected by a password. I don't know whether I missed the information or if it has not been provided to me.

ANSWER: We thank the Reviewer for spotting this and apologize for the oversight. The data have since been made publicly available and can be accessed through the following links (on **P15**):

<https://www.ncbi.nlm.nih.gov/geo/query/acc.cgi?acc=GSE283803> (scATAC-seq)

<https://www.ncbi.nlm.nih.gov/geo/query/acc.cgi?acc=GSE283804> (scRNA-seq)

<https://www.ebi.ac.uk/biostudies/ArrayExpress/studies/E-MTAB-14873> (WGBS)

Three populations of spermatogonia have been identified. It would be relevant to perform analyses on spermatogonia dynamics along with SSC self-renewal and differentiation as the dataset seem appropriate to do so. Indeed, the level of conservation of these processes across vertebrates is of high interest for the evo-devo community.

ANSWER: We thank the Reviewer for this suggestion. Indeed, this is a relevant analysis to perform especially as previous work has shown the existence of metastable states in human spermatogonia (Guo et al, 2018). To investigate potential differentiation trajectories, we employed RNA velocity analysis, a computational framework that utilizes the relative abundance of unspliced and spliced mRNA transcripts in scRNA-seq data to infer the prospective transcriptional states of individual cells. The ratio of unspliced to spliced reads serves as an approximation of ongoing transcriptional activity. By comparing these profiles to the steady-state transcriptomes of neighboring cells, a velocity vector can be computed for each cell, indicating both the direction and magnitude of its predicted transcriptional progression. These vectors are visualized within the UMAP embedding, offering insights into the dynamic transitions between cellular states. In our analysis, we observed prominent velocity vectors extending from undifferentiated spermatogonia (SPG-Aun) toward differentiated spermatogonia (SPG dif), aligning with the expected direction of developmental commitment. Notably, a subset of SPG-dif cells exhibited velocity vectors oriented toward the undifferentiated state, suggesting the presence of transcriptional plasticity and potential reversibility within this compartment. These observations indicate that both undifferentiated and differentiated SPG populations may contribute to the emergence of SPG B cells, reflecting a metastable and uncommitted cellular landscape, consistent with the dynamic plasticity previously reported by Guo et al. (2018) (<https://doi.org/10.1038/s41422-018-0099-2>). We have now added the following to the main text, whereas RNA velocity analysis is represented as **Appendix Figure S5**.

P3: “Given the critical role of spermatogonia in maintaining stem cell potential and initiating spermatogenic commitment, we next focused on delineating potential differentiation trajectories within this compartment. To this end, we employed RNA velocity analysis (La Manno et al, 2018) to infer both the direction and magnitude of predicted transcriptional progression across SPG populations. In our analysis, we observed prominent velocity vectors extending from undifferentiated spermatogonia (SPG-Aun) toward differentiated spermatogonia (SPG-Ad), aligning with the expected direction of developmental commitment. Notably, a subset of SPG-Ad cells exhibited velocity vectors oriented towards the undifferentiated state (SPG-Aun) (Appendix Fig. S5), suggesting transcriptional plasticity and potential reversibility within this compartment. These observations indicate that both undifferentiated and differentiated SPG populations may contribute to the emergence of SPG-B cells, reflecting a metastable and uncommitted cellular landscape, consistent with the dynamic plasticity previously reported in human spermatogonia populations (Guo et al., 2018).”

Minor points:

The quality of figure S5A is very low as it is not possible to read the labels.

ANSWER: We apologize for this. All the figures have now been provided in .pdf format in high resolution.

Why does the data come from transgenic zebrafish individuals?

ANSWER: We thank the Reviewer for spotting this. This is a mistake originating from simultaneous preparation of another manuscript which indeed employed transgenic zebrafish.

3) Additional metrics such as the percentage of intronic reads along spermatogenesis would be interesting as it reflects active transcription.

ANSWER: We have now added these data as **Appendix Figure S6** and commented on the findings in the main text.

P4: *“Moreover, we observed that the spermatogenesis process is paralleled by a gradual transcriptional shutdown, with elongated spermatids being almost entirely transcriptionally quiescent (Fig. EV1D). To better understand the RNA processing dynamics during spermatogenesis, we first examined the proportion of exonic, and intronic reads across germ cell populations (Appendix Fig. S6A). Intronic reads, which primarily represent unspliced pre-mRNA, can serve as a proxy for nascent transcription (La Manno et al., 2018). We found that the ratio of intronic to exonic reads remained relatively constant from spermatogonia to spermatids, suggesting a coordinated downregulation of both transcription and mRNA abundance without substantial accumulation of mature transcripts in later stages. We next quantified the absolute number of intronic reads per cell across biological replicates (Appendix Fig. S6B). This analysis revealed a progressive reduction in intronic read counts from early to late germ cell stages, consistent with a gradual decline in transcriptional activity during spermatid maturation (Fig. EV1D). Together, these findings support a model of transcriptional shutdown that occurs in a stepwise manner, likely involving both repression of transcription initiation and increased transcript turnover.”*

Reviewer #3:

Ana María Burgos Ruíz and colleagues developed a comprehensive single-cell multi-omics resource that combines single-cell RNA sequencing (scRNA-seq) and single-cell chromatin accessibility (scATAC-seq) profiling in zebrafish spermatogenesis with base-resolution DNA methylome (WGBS) analysis of sorted germ cell populations. scRNA-seq annotated the major germ cell populations: undifferentiated spermatogonia A (SPG-Aun), differentiated spermatogonia A (SPG-Ad), spermatogonia B (SPG-B), primary spermatocytes (SPC-I), secondary spermatocytes (SPC-II), round spermatids (SPT-r), and elongated spermatids (SPT-e), and identified a lot of new driver genes.

Next, to gain insight into gene regulation during spermatogenesis, we performed DNA methylation (5mCG) and found thousands of localized and bidirectional 5mCG changes in potential gene regulatory regions that occur at the spermatocyte stage. To further study the gene regulation mechanisms operating during spermatogenesis, they analyzed scATAC-seq libraries and found local chromatin changes associated with the activity of spermatogenic marker genes, as well as a stepwise reprogramming of global chromatin structure that leads to chromatin compaction and transcriptional shutdown.

Finally, they investigated the chromatin accessibility profile of elongated spermatids using scATAC-seq and found that late SPT-e ATAC-seq signals were indeed strongly consistent with placeholder chromatin, which was identified as a major driver of maternal-to-paternal chromatin remodeling before

ZGA. These findings are extremely useful for understanding the changes in chromosome structure accompanying changes in gene expression patterns during spermatogenesis, and provide valuable insight for understanding the chromosome structure of spermatozoa that transmit genome to the next generation in non-mammals.

ANSWER: We thank the Reviewer for the positive feedback and helpful suggestions to improve our manuscript

The results are comprehensive and have been tested in biological duplicate. Therefore, they are highly reliable, but I concern the number of SPG-Aun cells (Fig. 1D). In general, it is known that the number of germline stem cells, as well as undifferentiated germ cells, is overwhelmingly smaller than that of differentiated germ cells. In fact, Leal et al. (2009) reported that the number of undifferentiated germ cells was also small in histological observations. Authors should have an explanation for this difference.

ANSWER: We thank the Reviewer for raising this point. While our cell-type proportions are consistent with those reported in previous zebrafish scRNA-seq studies (Qian et al., Sposato et al.), they do differ significantly from estimates based on histological analyses. We believe this discrepancy arises from technical artifacts associated with droplet-based single-cell capture methods, such as those used by 10x Genomics. More resilient cell types like spermatogonia are preferentially recovered during enzymatic dissociation and microfluidic partitioning, whereas more fragile or structurally embedded cells such as spermatids tend to be underrepresented. We have now included this explanation in the revised manuscript text.

P4: *“We next assessed the total number of cells within each population and found comparable representation of major germ cell types across biological replicates (Fig. 1D), consistent with previous observations (Sposato et al., 2024). Nevertheless, these proportions differ from those reported in histological studies (Leal et al., 2009), which typically show spermatogonia as a minor population. The relative enrichment of spermatogonia in our scRNA-seq dataset likely reflects a known technical artifact; during enzymatic or mechanical tissue dissociation and microfluidic capture, more resilient cells, such as spermatogonia, are preferentially recovered, whereas more fragile or structurally embedded cells, such as spermatids, are often underrepresented. This phenomenon has been documented previously (Denisenko et al, 2020) and likely explains the distribution observed in our dataset.”*

It is also confusing that the notation of SPG is different for scRNA-seq, DNA methylome (WGBS) analysis, and scATAC-seq. The notation should be consistent throughout the paper, and if that is difficult, each should be clearly defined.

ANSWER: While our annotations are consistent throughout the manuscript, each performed assay offers a different level of resolution and may not fully capture the cellular complexity of spermatogenesis. For example, scRNA-seq allowed us to resolve seven major germ cell populations, as described on page 3.

P3: *“After manual curation of the clusters, which was based on expression of previously defined marker genes (Qian et al., 2022; Ye et al., 2023), we annotated major germ cell populations: undifferentiated spermatogonia-A (SPG-Aun), differentiated spermatogonia-A (SPG-Ad), spermatogonia B (SPG-B), primary spermatocytes (SPC-I), secondary spermatocytes (SPC-II), round spermatids (SPT-r), and elongated spermatids (SPT-e) (Fig. 1B and 1C; Fig. EV1C).”*

In contrast, for DNA methylation profiling via WGBS, which was performed on FACS-sorted germ cell populations, our resolution was limited by the availability of suitable markers, and we could not capture the full spectrum of cell types. We acknowledge that this does not provide the same level of resolution as scRNA-seq and have therefore clarified what is meant by "SPG" in this context.

P5: *“We obtained a mix of undifferentiated and differentiated spermatogonia (SPG), SPC-I, SPT-r populations, as well as mature sperm (SP) (Fig. 3B, C).”*

Similarly, for ATAC-seq, the populations defined mirror those of the scRNA-seq experiment, except for the spermatogonial compartment. In this case, we defined a population of differentiated spermatogonia (SPG-d), which comprises both SPG-Ad and SPG-B cells, as these subpopulations could not be resolved solely based on chromatin accessibility data. We have now clarified this point in the manuscript as follows:

P6: *“In contrast, differentiated spermatogonia (SPG-d; comprising a mixture of SPG-Ad and SPG-B cells) were identified by the absence of these markers and the activity of genes, including smad6, socs2, nop56, and zranb2, which were previously associated with later stages of spermatogonia and their progression (Guo et al., 2018; Itman & Loveland, 2008; Orwig et al, 2008; Shami et al., 2020).”*

Minor points

From the first line from the bottom of the left paragraph to the fourth line from the top of the right paragraph on page 4, please clearly indicate which populations are SPG populations and SPC cells.

ANSWER: This has now been clarified in the text.

P4: *“For example, SPG-Aun and SPG-Ad populations were enriched in terms associated with ribosome biogenesis and translation, further supporting the findings that transition from self-renewal to germline differentiation is dependent on ribosome biogenesis and increased protein synthesis (Sanchez et al, 2016). SPC-I cells were enriched in terms associated with meiosis and DNA repair, (Griswold, 2016), whereas SPT-r and SPT-e populations displayed enrichment in terms linked to cilium assembly (Mirvis et al, 2018).”*

The image quality of the supplemental figures is so low that Fig. S5A is almost unreadable.

ANSWER: We apologize for this. All the figures have now been provided in .pdf format in high resolution.

12th Sep 2025

Manuscript Number: MSB-2025-13014R

Title: A single-cell multiomics roadmap of zebrafish spermatogenesis reveals regulatory principles of male germline formation

Author: Ana Burgos-Ruiz

Fan-Suo Geng

Gala Pujol

Estefanía Sanabria-Reinoso

Thirsa Brethouwer

María Almuedo-Castillo

Aurora Ruiz-Herrera

Juan Tena

Ozren Bogdanovic

Dear Oz,

Thank you for sending us your revised manuscript. We have now heard back from the three reviewers who agreed to evaluate your revised study. As you will see below, the reviewers are satisfied with the performed revisions and support publication. Before we can proceed with formal acceptance, we kindly ask you to address the following remaining issue:

1. The final minor comment from Reviewer #3.

On a more editorial level, please address the following issues:

1. Please provide up to five keywords in the manuscript file.

2. Remove the "Author contribution" section from the manuscript file.

3. Ensure the funding information listed in the manuscript and entered in the submission system is consistent. Currently, the Spanish Ministry of Science and Innovation (AEI/10.13039/501100011033) is missing from the submission system.

4. Table EV1 and EV2 are too wide to be converted to PDF, therefore please rename them to EV Datasets and updated the nomenclature and callouts accordingly. Their legends should be removed from manuscript file but remain as separate tabs/sheets in each Excel file.

5. Ensure that all callouts are listed in sequential order. Rename the callouts for Appendix Tables 1-2 to Appendix Tables S1-S2. Additionally, include the missing callout for Table EV2 and rename it to Dataset EVx.

6. Please provide the synopsis image in PNG format instead of PDF, and adjust its dimensions to 550 pixels wide and between 400-600 pixels high.

7. Please provide a "standfirst text" summarizing the study in one or two sentences (approximately 250 characters, including space), three to four "bullet points" highlighting the main findings.

Here are a couple of examples:

<https://www.embopress.org/doi/10.15252/msb.20199356>

<https://www.embopress.org/doi/10.15252/msb.20209475>

<https://www.embopress.org/doi/10.15252/msb.209495>

8. Please address the following issues in figure legends:

- Please indicate the statistical test used for data analysis in the legends of figures 4C, EV1 E, EV5 A.

- Please note that information related to n is missing in the legend of figure 4D

Click on the link below to submit your revised paper.

Sincerely,
Jingyi

Jingyi Hou, PhD
Senior Editor
Molecular Systems Biology

*** PLEASE NOTE *** As part of the EMBO Press transparent editorial process initiative (see our Editorial at <https://dx.doi.org/10.1038/msb.2010.72> , Molecular Systems Biology will publish online a Review Process File to accompany accepted manuscripts. When preparing your letter of response, please be aware that in the event of acceptance, your cover letter/point-by-point document will be included as part of this File, which will be available to the scientific community. More information about this initiative is available in our Instructions to Authors. If you have any questions about this initiative, please contact the editorial office (msb@embo.org).

Reviewer #1:

The authors have address all the points raised in my review, as well as in the other reviews, and I believe this paper addresses an important question in germ cell biology and would be of interest to the scientific community and is suitable for publication

Reviewer #2:

The authors have addressed my concerns and the revisions have significantly improved the manuscript.

Reviewer #3:

I agree with the authors' revision. I have no further requests, but I would like to make one comment. The second line of the correction in red in the second paragraph on page 4:
"the findings that transition from self-renewal to germline differentiation is dependent on ribosome biogenesis and increased protein synthesis (Sanchez et al, 2016)."
Recently, it has been reported that upregulation of rRNA transcription and increased protein synthesis is required for spermatogonial stem cell differentiation in zebrafish (Kawasaki et al., 2025, eLife 14 RP104295). I think this fits better with this paper. However, there is no need to review this again. It is up to the authors' discretion whether to cite it or not.

All editorial and formatting issues were resolved by the authors.

22nd Sep 2025

Manuscript number: MSB-2025-13014RR

Title: A single-cell multiomics roadmap of zebrafish spermatogenesis reveals regulatory principles of male germline formation

Dear Dr. Bogdanovic,

Thank you again for sending us your revised manuscript. We are now satisfied with the modifications made and I am pleased to inform you that your paper has been accepted for publication.

Sincerely,
Jingyi

Jingyi Hou, PhD
Senior Editor
Molecular Systems Biology
